# Kinesin-2 and IFT-A act as a complex promoting nuclear localization of β-catenin during Wnt signalling

Linh T. Vuong[1], Carlo Iomini[2,3], Sophie Balmer[1,6], Davide Esposito[4], Stuart A. Aaronson[3,4,5] & Marek Mlodzik[1,3,5]

Wnt/Wg-signalling is critical signalling in all metazoans. Recent studies suggest that IFT-A proteins and Kinesin-2 modulate canonical Wnt/Wg-signalling independently of their ciliary role. Whether they function together in Wnt-signalling and their mechanistic role in the pathway remained unresolved. Here we demonstrate that Kinesin-2 and IFT-A proteins act as a complex during *Drosophila* Wg-signalling, affecting pathway activity in the same manner, interacting genetically and physically, and co-localizing with β-catenin, the mediator of Wnt/Wg-signalling on microtubules. Following pathway activation, Kinesin-2/IFT-A mutant cells exhibit high cytoplasmic β-catenin levels, yet fail to activate Wg-targets. In mutant tissues in both, *Drosophila* and mouse/MEFs, nuclear localization of β-catenin is markedly reduced. We demonstrate a conserved, motor-domain dependent function of the Kinesin-2/IFT-A complex in promoting nuclear translocation of β-catenin. We show that this is mediated by protecting β-catenin from a conserved cytoplasmic retention process, thus identifying a mechanism for Kinesin-2/IFT-A in Wnt-signalling that is independent of their ciliary role.

[1] Department of Cell, Developmental, and Regenerative Biology, Icahn School of Medicine at Mount Sinai, One Gustave L. Levy Place, New York, NY 10029, USA. [2] Department of Ophthalmology, Icahn School of Medicine at Mount Sinai, One Gustave L. Levy Place, New York, NY 10029, USA. [3] Graduate School of Biomedical Sciences, Icahn School of Medicine at Mount Sinai, One Gustave L. Levy Place, New York, NY 10029, USA. [4] Department of Oncological Sciences, Icahn School of Medicine at Mount Sinai, One Gustave L. Levy Place, New York, NY 10029, USA. [5] Tisch Cancer Institute, Icahn School of Medicine at Mount Sinai, One Gustave L. Levy Place, New York, NY 10029, USA. [6] Present address: Sloan Kettering Institute, New York, NY 10029, USA. Correspondence and requests for materials should be addressed to C.I. (email: carlo.iomini@mssm.edu) or to M.M. (email: marek.mlodzik@mssm.edu)

Canonical Wnt/Wingless signalling is a highly conserved pathway with important roles in the regulation of a variety of developmental processes, including cell fate specification, proliferation, cell survival, and migration[1–4]. Dysregulated expression of proteins in the Wnt/Wg pathway is often associated with diseases, including cancers[5–7]. Secreted Wnt proteins stabilize β-catenin (Armadillo/Arm in *Drosophila*), a multifunction protein that is critical in Wnt signaling as an essential transcriptional coactivator for Wg/Wnt-target gene expression[1–4]. Wnt/Wg proteins bind to Frizzled (Fz) receptors, resulting in alleviation of pathway inhibition caused by a "destruction complex", composed of APC (Adenomatous Polyposis Coli), Axin, GSK3β, and Casein kinase I (CKI), targeting β-catenin for degradation. Pathway activation results in relocalization of Axin to the plasma membrane[8] and stabilization of cytoplasmic β-catenin/Arm, which then translocates to the nucleus to act as coactivator of the Tcf/Lef transcription factors[1]. In the absence of Wg/Wnt, β-catenin/Arm is phosphorylated by the "destruction complex" and targeted to the proteasome[1,9,10].

*Drosophila* wing development serves as a paradigm for Wnt/Wg-signalling. In larval wing discs, Wg is expressed as a two-cell stripe at the dorso-ventral (D/V) boundary of the wing pouch, where it acts as a morphogen activating targets in a concentration-dependent manner[11–13]. Wg protein is detected in a gradient up to several cells away from the source[14], patterning wing development and specifying future wing margin structures[15,16]. High threshold targets of Wg, in cells adjacent to D/V boundary, include *senseless* (*sens*), a gene responsible for the formation of margin sensory bristles of adult wings[17], while lower threshold target genes include *Distalless* (*Dll*), required for wing growth and expressed in a graded fashion, decreasing towards the edges of the wing pouch[15,18].

Klp64D, *Drosophila* Kif3A, a KInesin-2 family member, and components of the IFT-A complex have recently been shown to be required for Wg signaling in non-ciliated *Drosophila* epithelial cells[19,20]. In absence of *Klp64D* function, Arm/β-catenin abnormally accumulates in cytoplasmic punctae, suggesting that Klp64D is involved in intracellular trafficking of Arm/β-catenin to enable Wnt/Wg signaling[19]. Klp64D/Kif3A is a subunit of the plus-end-directed microtubule-based motor Kinesin-2[21]. Kinesin-2 is a heterotrimeric holoenzyme consisting of two motor subunits, Kif3A and Kif3B, and the nonmotor subunit Kap3. Kinesin-2 drives the anterograde transport of IFT particles along the cillium[21,22]. In *Drosophila*, localization of Arm/β-catenin to adherens junctions (AJs) is also regulated by Klp64D in developing photoreceptor cells[23].

Intraflagellar Transport (IFT) is a movement of large protein complexes along axonemal microtubules, which is essential for the formation and maintenance of eukaryotic cilia and flagella. IFT particles are composed of about 20 proteins, organized into the IFT-A and IFT-B complexes[24,25]. Despite their essential role in ciliogenesis and cilia-associated signaling pathways in vertebrates, such as Hedgehog signaling[26], IFT-A proteins were also found to modulate canonical Wnt/Wg-signaling in non-ciliated cells in *Drosophila* imaginal discs[20], suggesting that IFT-A must function in the Wnt-pathway in a non-ciliary context. In cilia, IFT-A complexes control retrograde protein transport, from the tip to the base of the cillium[27–30]. Among the five conserved IFT-A proteins, four (IFT121, IFT122, IFT140, and IFT43) regulate Wnt/Wg-signaling in *Drosophila* through effects on β-catenin/Arm[20]. It remains unclear how this complex functions mechanistically in Wg/Wnt-signaling; whether IFT-A proteins associate with microtubular structures outside the cilium; and whether the non-ciliary Wnt-signaling-specific function of IFT-A is conserved in vertebrates.

The role of ciliary proteins in vertebrate Wnt-signaling has remained unresolved with existing data being confusing if not contradictory. Ciliary proteins have been suggested to limit response to Wnt signaling, affecting stability and localization of β-catenin/Arm[31]. There are many contradicting conclusions from analyses of Wnt signaling in the context of ciliary mutants, ranging from inactivation to hyperactivation of the pathway[31–33]. The barrier to understanding how ciliary proteins might function independently of the cilium stems from their crucial role in the biogenesis and maintenance of the cilium itself. It is thus difficult to distinguish cilia-dependent and independent effects in ciliated vertebrate cells. To overcome this problem we have used *Drosophila*, as all epithelial cells are non-ciliated in flies[34]. Consequently, phenotypes associated with loss of ciliary proteins must be due to cilia-independent function(s).

Here we analyze the cilia-independent function of Kinesin-2 and IFT-A complexes in Wg/Wnt-signaling. Our data indicate that Kinesin-2 and IFT-A interact via Kap3 to form a large complex and together are critical for nuclear translocation of Arm/β-catenin, and thus its activity in Wnt/Wg-signaling. We demonstrate that the combined Kinesin-2/IFT-A complex interacts directly with β-catenin/Arm through IFT140 and that this interaction is critical for normal nuclear translocation of β-catenin/Arm. Subcellular localization and the observation that kinesin motor activity is critical for the process suggest that the Kinesin-2/IFT-A complex acts via transport along cytoplasmic microtubules. Moreover, IFT140 interacts with a 50 amino acid region in Arm/β-catenin that is deleted in the stable and fully active mutant isoform ArmS10, suggesting that an inhibitor of nuclear translocation of Arm/β-catenin competes with IFT140. We thus identify a mechanism as to how the Kinesin-2/IFT-A complexes serve a critical function during Wnt/Wg-signaling, independently of their ciliary role, by stabilizing and promoting Arm/β-catenin translocation to the nucleus.

## Results

**Interaction of *Kinesin-2* and *IFT-A* in wing margin development.** Kinesin-2 and individual IFT-A protein components are essential for canonical Wg activity[19,20], and so we asked whether they associate with each other during Wg-signaling. We first tested this hypothesis by examining genetic interactions between single IFT-A proteins and knockdown of Klp64D (using the UAS/Gal4 system[35]). C96-Gal4 was used to drive expression at/near the D/V-boundary[36]. klp64D^RNAi under C96-Gal4 control (C96 > klp64D^RNAi) caused wing margin notching (Fig. 1a, b). Co-overexpression of IFT-A components, IFT121, IFT122, IFT140, and IFT143, suppressed the C96 > klp64D^RNAi phenotype in 100% of flies tested (Fig. 1c, d; Supplementary Figure 1), suggesting that the IFT-A complex acts together with Klp64D and that its components are rate limiting in knockdown backgrounds. We next confirmed this notion with molecular markers, examining whether interactions between IFT-A proteins and Klp64D affected expression of Wg-signaling targets *sensless* (*sens*) or *Distal-less* (*Dll*)[37,38]. Whereas C96 > klp64D^RNAi alone caused a marked reduction/loss in expression of both, Sens and Dll, in the D/V boundary region (Fig. 1b), co-overexpression of individual IFT-A components in C96 > klp64D^RNAi wing discs (e.g. C96 > klp64D^RNAi; >IFT122, C96 > klp64D^RNAi; >IFT140, C96 > klp64D^RNAi; >IFT121, or C96 > klp64D^RNAi; >IFT143) largely restored expression of Sens and Dll (Fig. 1c, d and Supplementary Figure 1). Consistently, coexpression of IFT144, which does not participate in Wg-signaling[20], failed to suppress the defects associated with C96 > klp64D^RNAi (Supplementary Figure 1d–e). These data suggest that Kinesin-2 and IFT-A components function together during Wg-signaling.

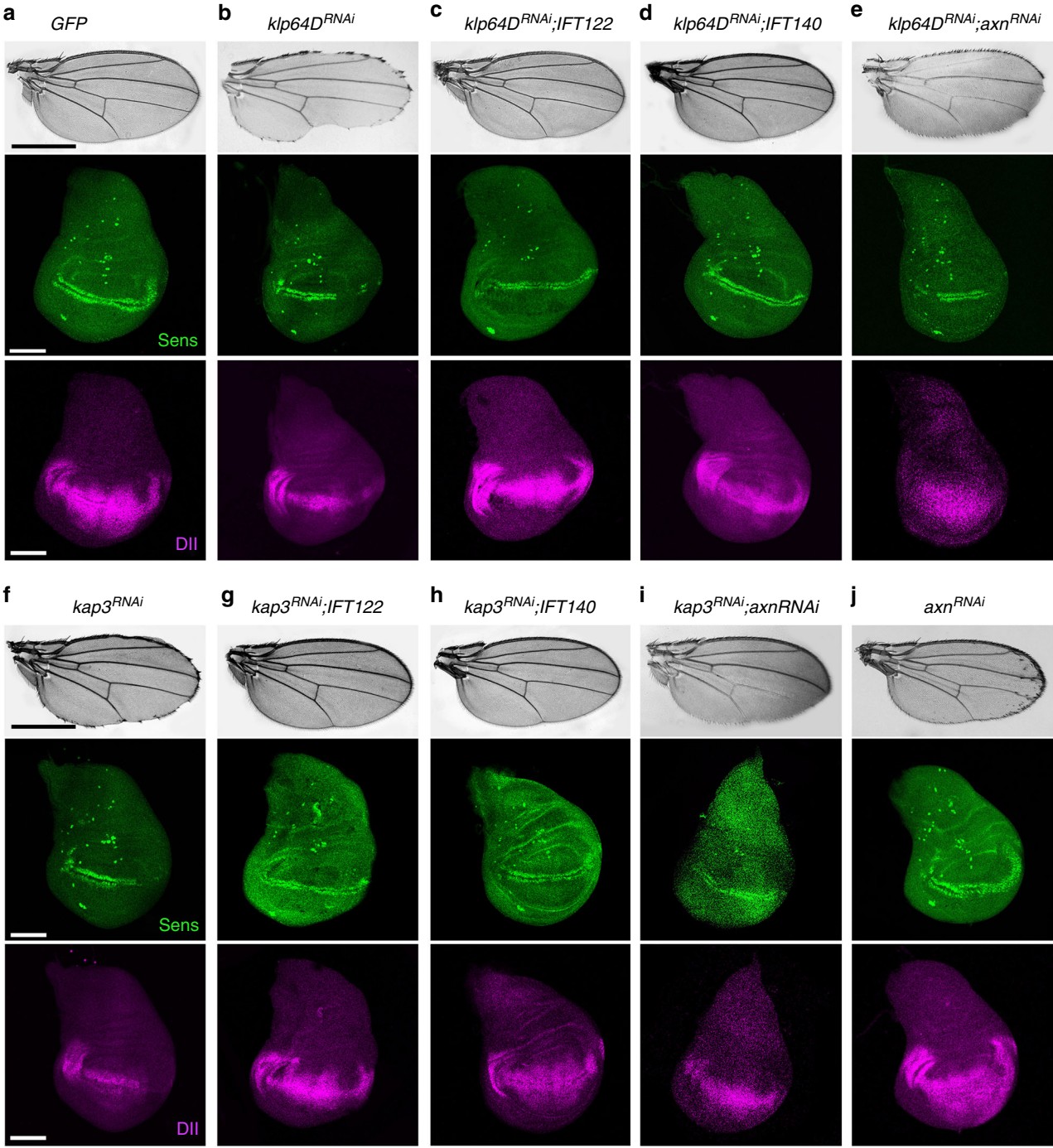

**Fig. 1** Genetic interactions between kinesin-2, IFT-A components, and Wg-signaling. All panels were using the wing margin driver *C96-Gal4*. Adult wings: anterior is up and distal to right; wing discs: ventral is up and anterior to the left. **a** *UAS-GFP* (*C96 > GFP*) control wing. *C96 > GFP* (control) with wild-type Sens (in green) and Dll (in magenta) expression at/near D/V boundary of wing imaginal discs. **b** *C96 > klp64D*^RNAi^, note wing notching phenotype. *C96 > klp64D*^RNAi^, note partial loss of Sens (in green) and reduction of Dll (in magenta), consistent with adult wing defects. **c, d** Notching caused by *klp64D*^RNAi^ is suppressed by co-overexpression of IFT-A components. Genotypes as indicated above panels. Sens (in green) and Dll expression (in magenta) were restored by co-overexpression of IFT-A components. **e** Notching caused by Klp64D knockdown is only partially suppressed *by axin*^RNAi^. Reduction of Sens and Dll expression (see b) was only partially restored by *axin* knockdown (see below, panel (**j**), for *axin*^RNAi^ alone as control). **f** *C96 > kap3*^RNAi^, note wing notching and margin loss. Reduced Kap3 function causes partial loss of Sens and Dll expression. Margin defects caused by *kap3*^RNAi^ knockdown were suppressed by co-overexpression of IFT-A components (**g**, **h**), and Sens and Dll expression at the D/V boundary in *C96 > kap3*^RNAi^ is restored by co-overexpression of IFT-A components. **i** Margin defects caused by *kap3*^RNAi^ knockdown are partially suppressed *by axin*^RNAi^. Reduction of Sens (green) and Dll expression (magenta) are also partially rescued by Axin KD. **j** *C96 > axin*^RNAi^, which hyperactivates the Wg pathway, induces extra margin bristles. *C96 > axin*^RNAi^ wing discs also show ectopic Sens and Dll expression. Scale bars represent 100 and 30 µm. All the tested wings show phenotype (*n* = 100)

Kif3A, the mammalian Klp64D homolog, functions in a heterotrimeric Kinesin-2 complex, including the motor protein (Kif3B) and a nonmotor accessory subunit (KAP3). In *Drosophila*, these subunits correspond to Klp64D, Klp68D, and Kap3, respectively[39]. Similar to *klp64D*[RNAi], RNAi-mediated knockdown of Kap3 (*C96 > kap3*[RNAi]) caused notching of adult wings and partial loss of Sens and Dll expression along the wing D/V-boundary (Fig. 1f). Coexpression of IFT122, and IFT140, in the *C96 > kap3*[RNAi] background suppressed the *C96 > kap3*[RNAi] wing phenotype and restored Sens and Dll expression (Fig. 1g, h), suggesting that the entire Kinesin-2 complex is involved in Wg-signaling together with IFT-A. While Kap3 reduction (*C96 > kap3*[RNAi]) affected wing development in a similar manner to Klp64D (Fig. 1b, f), knockdown of the third component, Klp68D, caused significantly smaller wings under the same conditions (Supplementary Figure 1f–h), suggesting that Klp68D may have Klp64D/Kap3-independent functions.

To confirm that the Kinesin-2/IFT-A complexes functioned in Wg-signaling, we tested whether Kinesin-2 depletion phenotypes could be rescued by Axin knock-down, a critical component of the Arm/β-catenin destruction complex and negative regulator of Wnt/Wg-signaling. Accordingly, single knockdown of Axin (*C96 > axn*[RNAi]) induced Wg-signaling gain-of-function phenotypes: ectopic margin bristles and Sens expression near the D/V boundary (Fig. 1j). Co-knockdown of *axn* with *klp64D* (*C96 > klp64D*[RNAi], *>axn*[RNAi]) or *kap3* (*C96 > kap3*[RNAi], *>axn*[RNAi]) partially suppressed the single *axn* knockdown phenotype, suggesting that *kap3* and *klp64D* are required downstream of Axin and might act on stable Arm/β-catenin released from the destruction complex (Fig. 1e−i). The double knockdowns resembled the single *kap3*[RNAi] and *klp64D*[RNAi] phenotypes, consistent with them acting downstream of *axn* (Fig. 1e−i). Taken together with the IFT-A analyses[20], these data indicate that the Kinesin-2 and IFT-A complexes function together in Wg-signaling-mediated wing development.

**Kinesin-2/IFT-A is required for Wg-target gene activation.** To corroborate where in the Wg/Wnt-pathway these complexes act, we examined double mutant clones of Kinesin-2 and canonical Wg-pathway components, such as Axin. First, we confirmed that Sens expression and Arm levels were lost or reduced, respectively, within *ift122*[179] and *klp64D*[k1] single mutant clones (Fig. 2a, b; loss of these markers is not due to cell death or cellular architecture, as junctional markers like Discs-large/Dlg were normally expressed (Supplementary Figure 2a–b, also ref. [20]). Similarly, double mutant clones for *ift122*[179]; *klp64D*[k1] (marked by absence of GFP, Fig. 2c) displayed indistinguishable phenotypes to single mutants, loss of Sens (Fig. 2c) and a reduction in Arm levels (Fig. 2c, Supplementary Figure 2d, see also below), confirming that a Kinesin-2/IFT-A complex affects Wg-signaling.

To test the hypothesis that Kinesin-2/IFT-A act downstream of Axin (Fig. 1; also ref. [20]), we analyzed target gene expression in double mutant clones of *Klp64D* and *axin*. Single *axin* mutant clones displayed increased Arm levels and ectopic Sens expression near the wing D/V boundary (Fig. 2d). Strikingly, double mutant clones of *klp64D*[k1] and *axn*[E77] displayed a loss of Sens expression at the D/V boundary (Fig. 2e: *klp64D*[k1] clones marked by absence of GFP, and *axn*[E77] clones by lack of RFP; double mutant clones lacked both markers, RFP and GFP; compared to single mutant *axn*[E77] clone left of the double mutant clone), although Arm/β-catenin levels were still increased as in *axn*[E77] single clones (Fig. 2e, compare single and double mutant area in twin clone marked by arrowheads). Note that cell junctions stained with Dlg in mutant cells appeared intact and indistinguishable from wild-type cells outside the clones (Supplementary Figure 2e). These results

support the notion that Kinesin-2/IFT-A act together, as a complex, and indicate that they are required downstream of Arm stabilization in Wg/Wnt-signaling, raising the possibility that the Kinesin-2/IFT-A complexes are required to promote translocation of stabilized Arm from the cytoplasm to the nucleus, where it activates target gene expression.

**IFT-A interact physically with Kinesin-2 and Arm/β-catenin.** The in vivo studies suggest that Kinesin-2/IFT-A complexes, and Arm/β-catenin might physically interact. We tested this biochemically via co-immunoprecipitation (co-IP) from in vivo samples. Proteins extracted from third instar wing imaginal discs of *tub > IFT122-GFP* or *tub > IFT140-GFP* larvae were immunoprecipitated with anti-GFP and analyzed by immunoblotting with anti-Arm antibody. Endogenous Arm/β-catenin is immunoprecipitated with IFT-A components (Fig. 3a; control genotypes of *tub > GFP* or *tub > IFT144* did not precipitate Arm). Their association was confirmed by direct binding in in vitro assays. IFT140 was sufficient to pull down Arm/β-catenin (Fig. 3b). Interactions between Klp64D with IFT122/IFT140 were also detected by in vivo co-IPs from *tub > IFT122-GFP* and *tub > IFT140-GFP* wing disc extracts (Fig. 3c). We tested in vitro which subunit of Kinesin-2, Klp64D or Kap3, bound directly to IFT-A and detected that IFT140 bound Kap3, but not Klp64D (Fig. 3d, e). As negative controls we used another kinesin, Klp61F (belonging to the human kinesin-5 family[40]) that is not involved in Wg-signaling (*klp61F*[RNAi] knockdown had no effect on wing development; Supplementary Figure 3h) and the IFT-A complex protein IFT144, which does not participate in Wg-signaling. These results defined that Klp64D/Kinesin-2 and IFT-A components are associated as a multiprotein complex, which interacts directly with Arm/β-catenin via IFT140 (Fig. 3f for architecture of the complex). Interestingly, Axin can also be found in a complex with IFT-A proteins (Supplementary Figure 3), but this is not followed up here further. The architecture of the physical interaction model of these complexes (Fig. 3f) was confirmed by knockdown of IFT-A components or Kap3 individually in wing discs and examined how loss of one component affected complex composition (Supplementary Figure 3d–f). However, in loss of IFT-A components or Kap3 the interaction between Klp64D and Arm was maintained (Supplementary Figure 3g–i), consistent with data that Klp64D can also bind Arm directly[19]. The data obtained using in vivo co-IP and pull-downs, in conjunction with the genetic associations, favor the conclusion that Arm/β-catenin exists in an IFT-A/Kinesin-2 complex architecture after Wg stimulation (also Fig. 3f).

We thus next asked whether these proteins colocalize in situ and whether colocalization was Wg dependent. As there are no functional IFT-A and Klp64D antibodies for in vivo staining, we expressed tagged Klp64D-HA with IFT140-myc, and stained with antibodies to the respective tags and Arm. To be able to analyze subcellular localization of Arm with sufficient resolution, we established a Wg-signaling assay in vivo in the large cells of the salivary glands (using a salivary gland specific *Gal4* driver; see below in next section for assay description). Confocal microscopy of salivary glands revealed intracellular puncta costained for Klp64D-HA and IFT140-myc (Fig. 4a, e). In absence of Wg expression (Supplementary Figure 4a), Arm/β-catenin is detected only at the adherens junctions (AJs) of membranes (Fig. 4a, also quantified in Fig. 4i). When Wg was expressed in salivary glands (Supplementary Figure 4a), Arm was detected both at the AJs and in puncta in the cytoplasm (Fig. 4b). Strikingly, many Arm/β-catenin, Klp64D, and IFT140 triple positive puncta were detected (Fig. 4b, f, i; note that these puncta often appear as "beads on a string", suggesting filament association). To establish whether

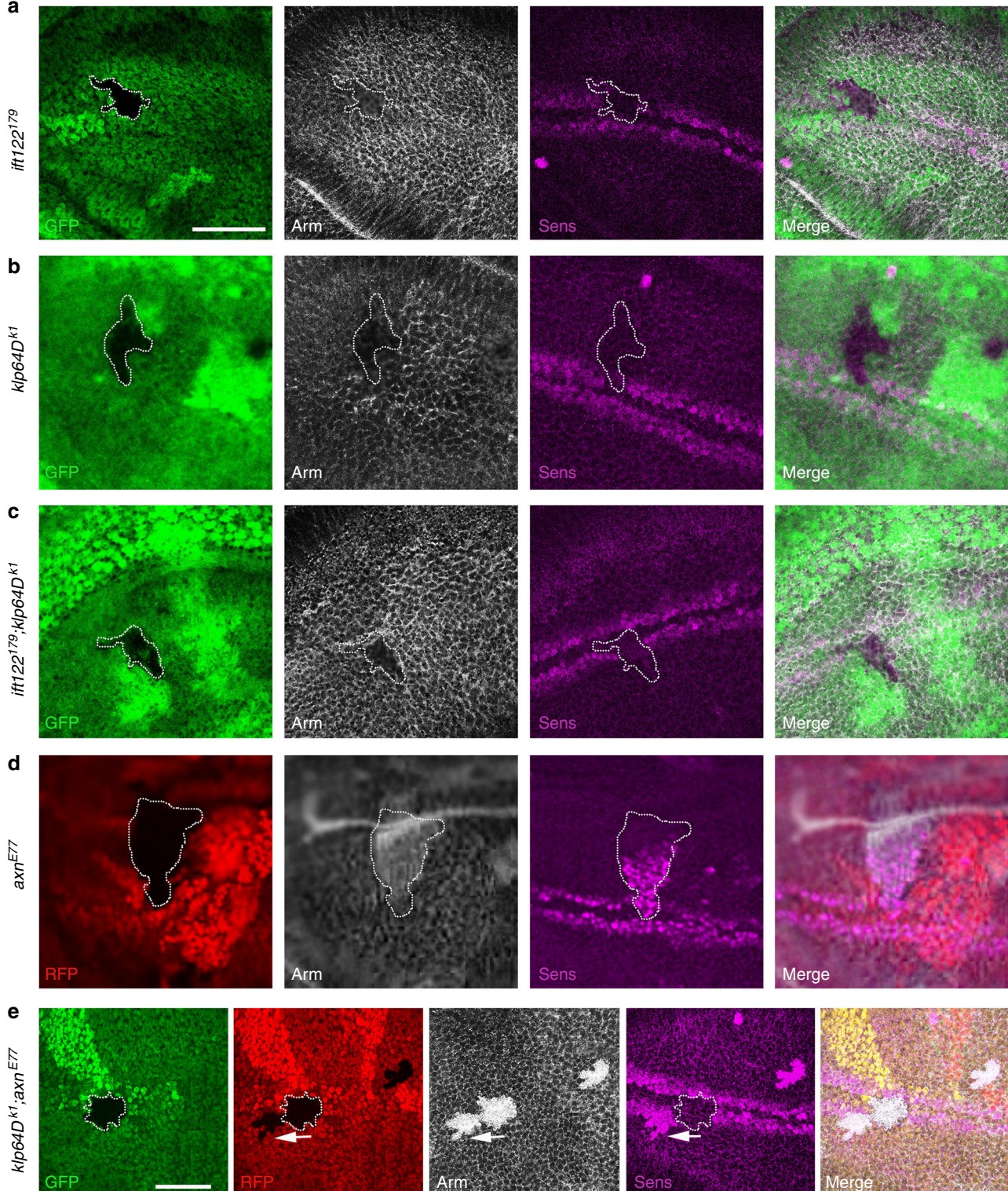

**Fig. 2** The kinesin-2/IFT-A complex is required for Wg target gene expression. All panels show wing discs near wing margin (evident by Sens expression, magenta), dorsal is up. Mutant clones (*n* = 20 clones in ten different wing discs) are marked by absence of GFP (green), RFP (red), or both; staining is with Arm (white), and Sens (magenta) as indicated. Genotypes of clones are as labeled on left. **a** Expression of Wg signaling target Sens is reduced or lost in *IFT122* mutant cells. Cytoplasmic Arm levels are reduced in *IFT122* mutant cells, and *IFT122* mutant cells display loss or reduction of Sens. **b** Wg target Sens is reduced in *klp64D* mutant cells. Cytoplasmic Arm levels are reduced in *klp64D* mutant cells, and Sens expression is lost. **c** Expression of Sens is lost in *ift122, klp64D* double mutant clones. Cytoplasmic Arm levels are markedly reduced in double mutant cells, and mutant cells display loss of Sens. **d** Ectopic expression of Wg target Sens in *axin* mutant clones. Cytoplasmic Arm levels are increased in *axin* mutant cells, and mutant cells display ectopic Sens expression away from margin. **e** *klp64D, axin* double mutant clones (*klp64D* marked by absence of GFP and *axin* RFP, double mutant clones lack both markers, outlined with white line). Cytoplasmic Arm levels are increased in both single and double mutant clones, but double mutant cells display loss of Sens expression (note example of *axin* single mutant clone lacking only RFP, marked by arrow). Scale bars represent 50 and 30 μm

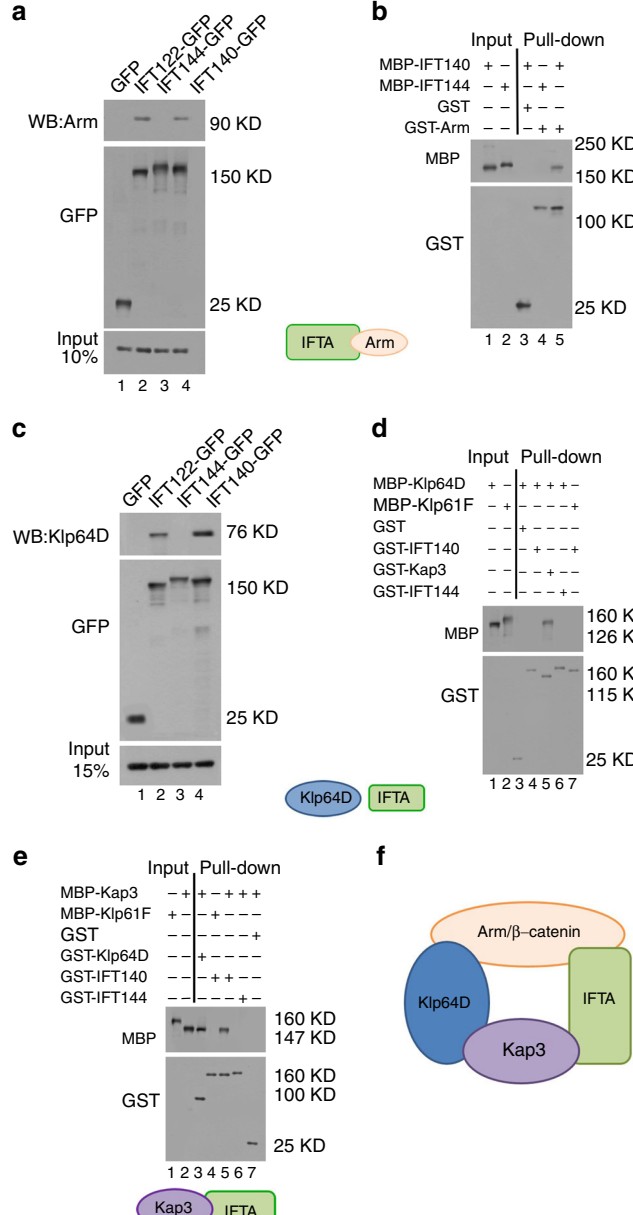

**Fig. 3** Kinesin-2, IFT-A, and Arm/β-catenin are physically associated. **a, c** Co-immunoprecipitation assay of IFT-A components with Klp64D or Arm from *tub > IFT122GFP* (lane 2) and *tub > IFT140GFP* (lane 4) or *tub > IFT144* (as a negative control, lane 3) wing imaginal discs. Protein extracts from wing discs were immunoprecipitated (IPed) with anti-GFP. IPs and input (10−15% of wing disc lysates used in IP step) were analyzed by blotting with antibodies to Arm (**a**), or Klp64D (**c**). Note that IFT-A components (IFT122 and IFT140) associate with Arm/β-catenin (**a**) and Klp64D (**c**). **b** Direct binding of IFT140 and Arm. Full-length IFT140 (lane 1: input 15%) was pulled down by GST-Arm (lane 5), IFT144 and GST alone are negative control (lanes 3, 4). **d** Klp64D is indirectly associated with IFT140. Full-length Klp64D (lane 1, 10% input) was not pulled down by GST (lane 3, negative control), or GST-IFT144 (lane 6, negative control), or GST-IFT140 (lane 4), but was pulled down by GST-Kap3 (lane 5; positive control). Full-length Klp61F (negative control; lane 2: 10% input) was not pulled down by GST-IFT140 (lane 7). **e** IFT140 directly binds to Kap3. Full-length Kap3 (lane 2: 10% input) was pulled down by GST-Klp64D (lane 3; positive control) and GST-IFT140 (lane 5), but not by GST-IFT144 or GST (lanes 6, 7; negative control) or full-length Klp61F (negative control; lane 1: 10% input; lane 4, pull-down assay). **f** The physical interaction model between Kinesin-2, IFT-A components, and Arm/β-catenin

colocalization between Klp64D/IFT140 and Arm also overlaps with cytoskeletal proteins, we tested whether IFT140 and Arm localization overlapped with α-tubulin. In absence of Wg, only IFT140 and α-tubulin are detected to overlap (Fig. 4c, g, i). However, in the presence of Wg expression, many triple positive α-tubulin, IFT140, and Arm punctae were observed to co-stain, again in a "string"-type arrangement in the cytoplasm of salivary glands, consistent with Arm being tethered to microtubules (Fig. 4d, h, i). We also confirmed colocalization of Klp64D-HA, IFT140-myc, and Arm in wing imaginal disc cells near the margin, in cells that are responding to Wg (Supplementary Figure 4). These results suggest that a significant fraction of Klp64D, IFT140 and Arm/β-catenin colocalize together on microtubules in the same punctate structures upon Wg-signaling activation. Together with functional in vivo and binding studies, these data indicate that the association of Arm/β-catenin with the Kinesin-2/IFT-A complex is important for Wg signaling.

**Kinesin-2/IFT-A is required for nuclear localization of Arm.**
Our data suggest that the Kinesin-2/IFT-A complex is required for Wg-signaling downstream of the destruction complex and thus affect the function and/or localization of Arm/β-catenin. In double mutant $klp64D^{k1}$, $axn^{E77}$ clones (Fig. 2; or also $ift140/rempA^{21Ci}$ and $axn^{E77}$; ref. [20]) Arm levels are high, indistinguishable from single $axn^-$ clones. However, loss of Sens expression in these double mutant clones suggested that the Kinesin-2/IFT-A complex is required for Arm/β-catenin function downstream of the destruction complex, possibly in the process of translocation to the nucleus.

It is very difficult, if not impossible, to document nuclear Arm in imaginal disc tissues due to the small size of cells. We therefore turned to the salivary gland assay mentioned above (Fig. 4), with its large cells, to address whether and how Kinesin-2/IFT-A complexes might affect subcellular Arm/β-catenin localization or function during Wg-signaling. To establish salivary glands as a useful system, we first tested whether expression of Wg there caused pathway activation-associated increase in cytoplasmic Arm/β-catenin levels and its nuclear translocation. Wg was expressed via the salivary gland-specific driver *C135-Gal4*/or *C805-Gal4*[41] to activate the pathway and anti-Arm staining used to analyze its levels and localization (Fig. 5a, *Drosophila* E-cadherin/DE-cad and Hoechst were used as cell membrane and nuclear markers, respectively). To facilitate detection of potential nuclear Arm/β-catenin, the salivary glands were also treated with Leptomycin B (LepB, a CRM1 inhibitor, which blocks nuclear export) before fixation and detection (as control we used expression of Exd-V5, which is involved in nuclear shuttling and known to remain inside the nucleus after LepB treatment;[42] Supplementary Figure 5a–b). Similarly, as second positive assay control, expression of ArmS10, a constitutively active and stable form of Arm/β-catenin, revealed Arm/β-catenin that was mainly localized to the nucleus in the presence of LepB (Supplementary Figure 5d), whereas ArmS10 expression without LepB displayed low levels of nuclear ArmS10 only in few cells (Supplementary Figure 5c), suggesting that Arm is actively exported from the nucleus without LepB treatment. Importantly, exogenous expression of Wg to activate the pathway revealed increased levels of Arm/β-catenin, detected in both nuclei and cytoplasm, indicating that Wg-induced Arm stabilization and nuclear translocation took place in salivary gland cells (Fig. 5a; note unaffected DE-cad staining, as control and cellular outline). As expected, expression of Wg without LepB treatment did not reveal detectable levels of nuclear Arm (Supplementary Figure 5f). Of note, the increase in Arm/β-catenin level was variable from cell to cell (Fig. 5 and Supplementary Figure 5), likely due to varying expression levels

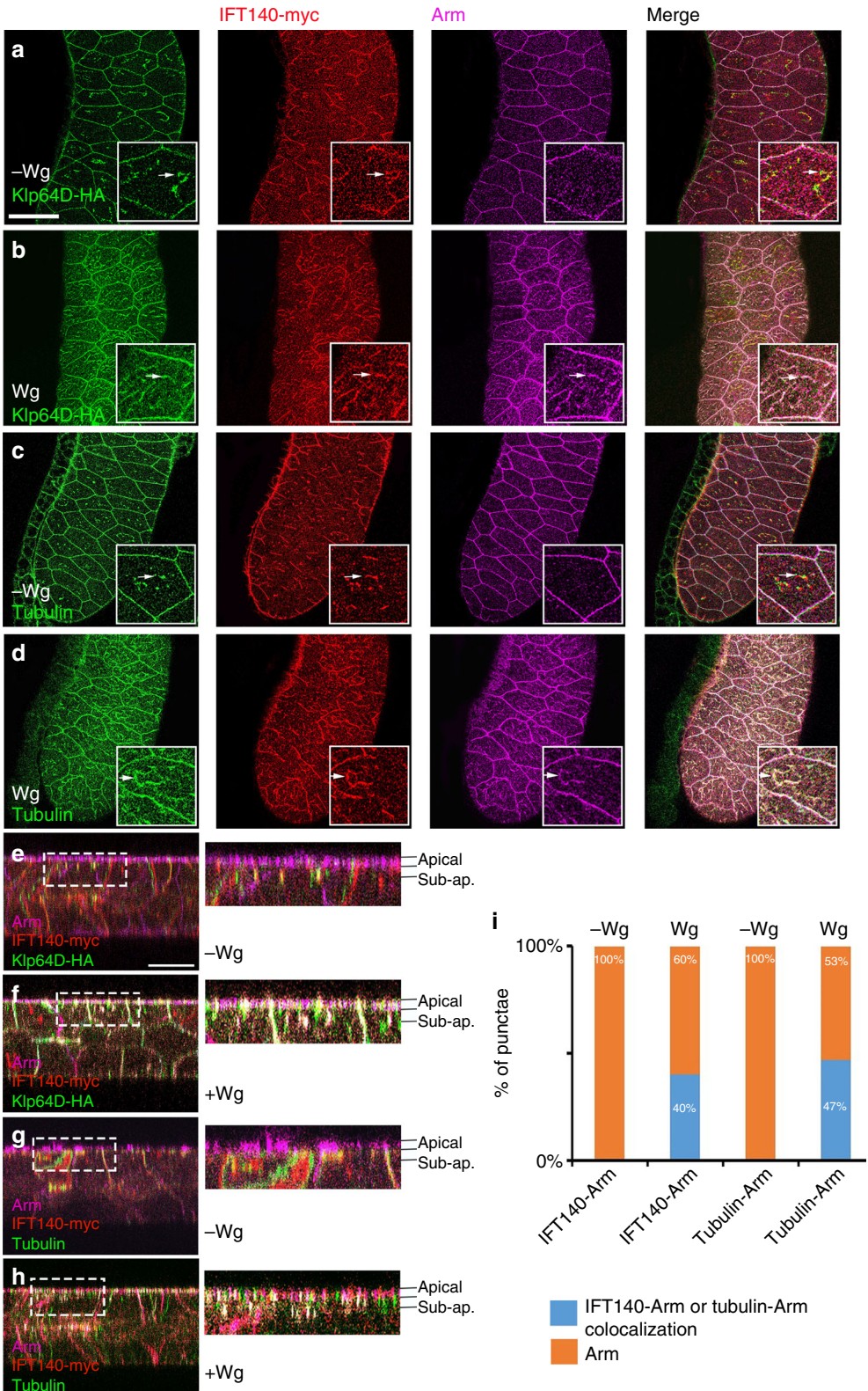

of the *Gal4*-driver. After establishing these parameters for the "Wg-signaling salivary gland assay", we used the *C805-Gal4 > Wg* background with LepB treatment and tested for effects of individual Kinesin-2/IFT-A complex RNAi-based knockdowns (KDs) on Arm/β-catenin localization.

KD of *ift140* and *klp64D* revealed a striking loss of nuclear Arm staining in this Wg-signaling assay (Fig. 5b, c), suggesting that the

Kinesin-2/IFT-A complex is required for nuclear translocation. To ask whether motility along microtubules (MTs), as mediated by Klp64D, was required, we tested whether a mutant motor domain of Klp64D could affect Arm localization in the salivary gland assay. We employed two kinds of mutant proteins that both bind MTs and adapters/cargo (like IFT-A) but fail to move along MTs: Klp64D$^{\Delta ABD}$, a deletion of the eight amino acid core

**Fig. 4** Wg-induced colocalization of Arm/β-catenin and kinesin-2/IFT-A complex. **a**, **b** Salivary glands were stained for Klp64D-HA (green), IFT140-myc (red), and Arm (purple) ($n = 30$). Punctate staining of Klp64D-HA overlaps with IFT140-myc (example marked by white arrow in insets). Insets are higher magnifications of salivary gland cells in panels (**a**–**d**). Without expression of Wg, Arm/β-catenin mainly localizes to junctional regions at the membrane (**a**), while upon Wg (**b**) expression Arm/β-catenin is stabilized (note increase in Arm levels, compare to panel (**a**)) and triple positive puncta staining (example marked by arrows in the insets) of Klp64D-HA, IFT140-myc, and Arm are detected in the cytoplasm. **c**, **d** Tubulin staining colocalizes with IFT140-myc. Without expression of Wg (**c**), Arm/β-catenin mainly localizes to the junctional regions of the membrane, while upon Wg induction Arm/β-catenin is also detected in triple positive puncta of β-Tubulin, IFT140-myc, and Arm in the cytoplasm (also note increase in Arm levels). **e**–**h** XZ-sections of salivary glands, stained as in **a**–**d**, with **e** and **g** being without Wg expression, and **f** and **h** upon Wg induction. Note punctate staining and overlapping localization at the sub-apical region of salivary gland cells containing Arm; in **f** and **h**. Boxed areas are shown at higher magnification on right. Scale bar in **e**, serves all micrographs except enlargements, represents 50 μm. **i** Quantification of the number of punctae containing Arm alone (orange) or Arm with IFT140 or tubulin (blue, $n > 200$ punctae per genotype, from five different salivary glands)

residues in the ATP binding pocket (aa107−114), and Klp64D$^{T114A}$, a Thr-to-Ala transition at T114 which blocks ATP binding[19] (these mutants share all functions of wild-type Klp64D except its capability to move along MTs). Strikingly, expression of either klp64D$^{ΔABD}$ or klp64D$^{T114A}$ resulted in a loss of nuclear Arm/β-catenin localization, and abnormally high cytoplasmic Arm levels as expected by Wg-signaling activation (Fig. 5d, e; note that cell membrane DE-cad staining was normal, Fig. 5d, e). These data suggest the importance of motor activity of the Kinesin-2 complex for Arm nuclear translocation in Wg signaling.

To corroborate this, we also activated the Wg-pathway at the level of the destruction/Axin complex. According to functional studies in wing discs, axin$^{RNAi}$ displayed highly enriched Arm/β-catenin staining, which was with LepB treatment mainly detected in the nucleus (Fig. 5f; note that without LepB treatment there was very little detectable nuclear Arm/β-catenin, even in a hyperactivated genotype like C805-Gal4 > Wg, > axin$^{RNAi}$; Supplementary Figure 5i). As seen with Wg-activated signaling, in double mutant cells with both reduced axin$^{RNAi}$ and a KD of either ift140 or Klp64D, an increased (stabilized) Arm/β-catenin level was detected, but it was constrained to the cytoplasm even in presence of LepB treatment (Fig. 5g, h; note intact cell membranes stained with DE-cad, Fig. 5g, h; see Supplementary Figure 5j–k for same genotypes without LepB treatment).

In summary, these data indicate that Kinesin-2/IFT-A complexes are required for nuclear translocation of Arm/β-catenin via movement along MTs, downstream of its release from the "destruction complex" following activation of Wg-signaling.

**Role of N-terminal Arm region in nuclear translocation.** Upon Wnt/Wg presence, the destruction complex is inhibited, resulting in the stabilization of Arm/β-catenin in the cytosol, so that it can then enter the nucleus to activate transcription of Wg/Wnt target genes[1–4]. To address the nuclear translocation feature further in vivo, we used the ArmS10 mutant isoform transgene, as overexpression of wt-Arm does not cause detectable defects. ArmS10 is a stable, constitutively active Arm-isoform, generated by a deletion of 54 amino acids near its N-terminus[43], a region normally targeted by phosphorylation of the "destruction complex". Accordingly, C96-Gal4 > ArmS10 induced a Wg-signaling gain-of-function (GOF) effect in wings with extra margin bristles and associated increase in cells expressing Sens within the Gal4-driver domain along the D/V boundary (Fig. 6a, c). In the nuclear translocation assay in salivary glands (with LepB treatment), ArmS10 localized predominantly to the nucleus as expected (Fig. 6h, cf. to Wg-activated endogenous Arm, Fig. 6a; also Supplementary Figure 5d). Strikingly, while C96-Gal4 > klp64DRNAi caused a wing notching phenotype and loss of Sens expression as expected (Fig. 6b; cf. Fig. 1) and eliminated

detectable nuclear translocation in salivary gland cells (Fig. 6g, with LepB treatment; cf. Fig. 5), it had no effect on ArmS10-caused wing phenotypes or its nuclear translocation in salivary glands (Fig. 6d, i). Similarly, in ift140 KD-backgrounds the GOF phenotype of ArmS10 and its nuclear localization in salivary glands was not affected (Fig. 6e, j; quantifications in Supplementary Figure 6). This is in stark contrast to the functional requirement of the Kinesin-2/IFT-A complex in nuclear translocation of stabilized endogenous Arm/β-catenin (Figs. 2e and 5).

To gain mechanistic insight into this striking discrepancy, we used a differently stabilized, constitutively active Arm/β-catenin isoform, Arm* 9 (ref. [44]). The difference between ArmS10 and Arm* is that the phosphorylation region, targeting it for degradation, is deleted in ArmS10, while Arm* carries a non-phosphorylable mutation in the CK1 phospho-target site, T52A (schematized in Fig. 6k, refs. [43,44]). Both are stable, because they cannot be phosphorylated by destruction complex kinases[43,44]. Expression of Arm* with C96-Gal4 has a similar effect on wing development as ArmS10 with ectopic margin bristles and increase in cells expressing Sens (Fig. 6l)[43]. In contrast to ArmS10, simultaneous knockdown of klp64D or ift140 reversed the Arm* GOF phenotype and Sens expression back to near wild-type patterns (Fig. 6m, n; quantified in Supplementary Figure 6). Consistently, in the salivary gland assay, Arm* was mainly localized to the nucleus (Fig. 6o; with LepB treatment), similarly to axn$^−$ mutants or ArmS10 expression (Figs. 5f, 6h). However, in contrast to ArmS10, a KD of Kinesin-2/IFT-A complex components in the Arm* background caused Arm* to be localized to the cytoplasm, like endogenous Arm (Fig. 6p, q; with LepB treatment). This scenario, stabilized yet trapped in the cytoplasm, was similar to stable endogenous Arm, generated via axn-KD in klp64D or ift140 KD-backgrounds, indicating that without the function of Kinesin-2/IFT-A stable full-length Arm is "prevented" from translocating efficiently to the nucleus upon Wg-signaling activation.

The differential behavior of endogenous Arm/Arm* vis-à-vis Arm S10, identified ArmS10 as a functional outlier and raised the possibility that the Kinesin-2/IFT-A complex might interact directly with the small region deleted in ArmS10 (residues 34 −87). We tested this hypothesis directly in in vitro pull-down experiments between IFT140 and Arm34-87 using purified recombinant GST-fusion proteins. Whereas Klp64D or IFT144, as negative controls, did not pull-down Arm34-87, IFT140 displayed strong binding to Arm34-87 (Fig. 7a), indistinguishable from full-length Arm (Fig. 3b), indicating a direct interaction between IFT140 and this Arm region.

To confirm this in vivo, we performed co-IP assays with wing disc extracts between Klp64D or IFT140 and Arm* (full length) or ArmS10. Extracts from C96 > IFT40-GFP and C96 > ArmS10; >IFT140-GFP were immunoprecipitated using anti-GFP and probed with anti-Arm (Fig. 7b), revealing that ArmS10 did not co-IP with IFT140 (Fig. 7b; the only detected Arm protein is

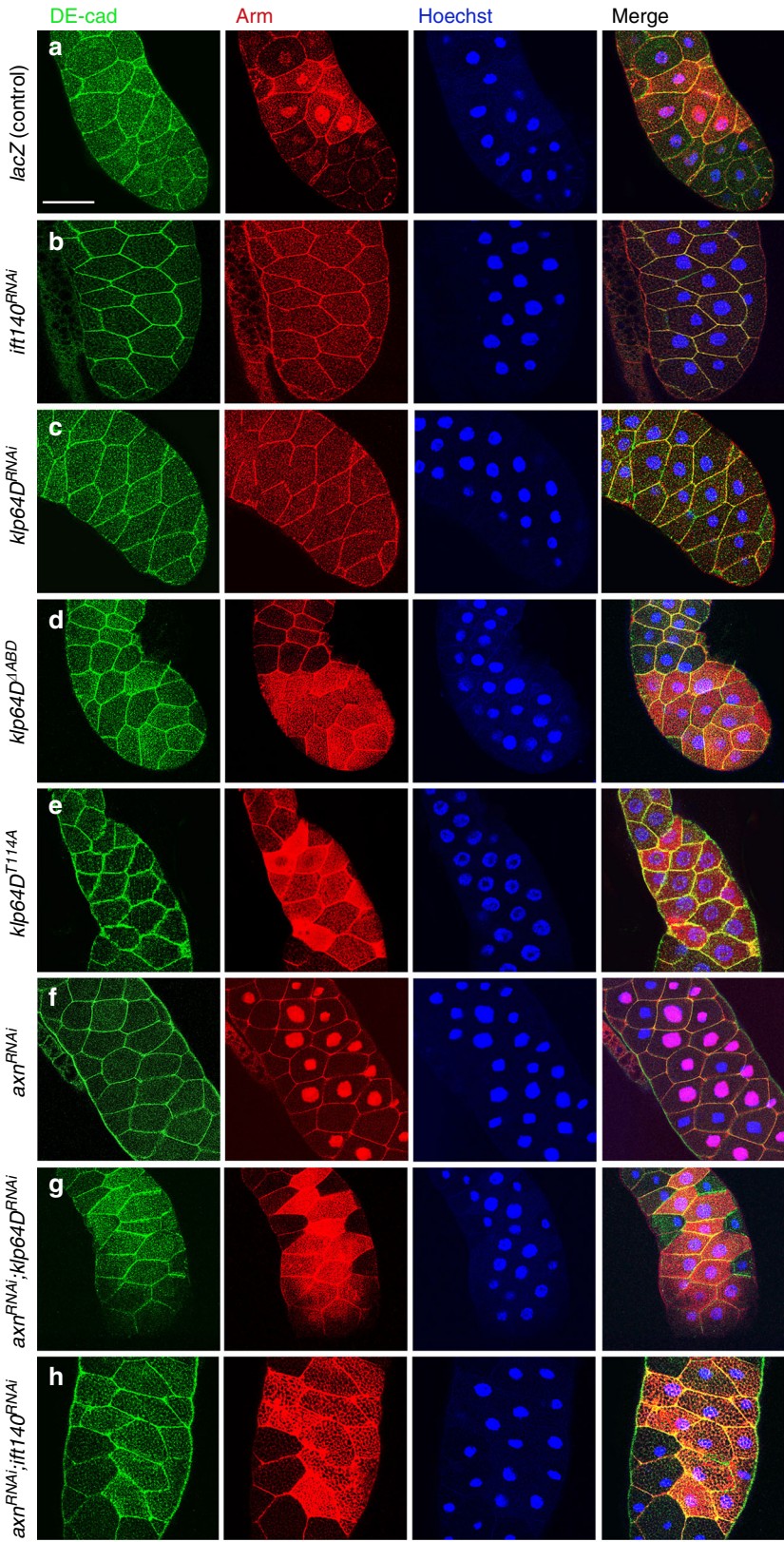

endogenous Arm, which served as positive control). In contrast, testing Arm* in the same assay revealed a robust interaction/co-IP between Arm* and IFT140 (Fig. 7c).

Together, these data indicate that association of the Kinesin-2/IFT-A complex with Arm/β-catenin is required for efficient Wg-signaling by directly interacting with Arm/β-catenin and promoting its nuclear translocation to activate Wg target genes. As ArmS10 can translocate to the nucleus independent of Kinesin-2/IFT-A and Arm/β-catenin directly bound to IFT140 via the region deleted in ArmS10, these data indicate that Arm/β-catenin can interact with another factor/complex that inhibits its nuclear translocation. This means that the Kinesin-2/IFT-A

**Fig. 5** Arm/β-catenin nuclear localization depends on kinesin-2/IFT-A complex function. Salivary glands were exposed to Wg expression (via *C805-Gal4* driver) and treated with Leptomycin B (LepB, a CRM1 inhibitor, which largely blocks nuclear export) to enhance retention of Arm/β-Catenin in the nucleus. Salivary glands were stained with DE-cad (green, as control and membrane marker), Arm (red), and Hoechst (nuclei, blue). All panels are *C805-Gal4 > Wg* and as indicated: **a** *UAS-lacz* (*>Wg, >LacZ*; control), note Arm/β-Catenin localization in both cytoplasm and nucleus (patchy increase in Arm/β-Catenin level is caused by uneven expression of Gal4-driver). **b** *>Wg, >ift140^RNAi*: Arm/β-Catenin nuclear localization is largely lost. **c** *>Wg, >klp64D^RNAi*: Arm nuclear localization is lost. **d** *>Wg, klp64D^ΔABD*: Increased cytoplasmic Arm/β-Catenin levels are observed, but nuclear staining is not detected. **e** *>Wg, klp64D^T114*: Arm levels are increased in the cytoplasm, but not present in the nucleus. **f** *>Wg, >axn^RNAi*: high Arm/β-Catenin levels and nuclear localization are detected in the *axn* knockdown background. **g** *>Wg, >axn^RNAi, >klp64D^RNAi*: Increased Arm/β-Catenin levels are detected, but largely restricted to the cytoplasm, and nuclear localization is markedly reduced. **h** *>Wg, >axn^RNAi, > ift140^RNAi*: Again, increased cytoplasmic Arm/β-Catenin levels are detected, but nuclear Arm/β-Catenin is largely lost. Scale bar represents 50 μm. The phenotype is observed in all experimental salivary glands (*n* ≥ 20)

complex is required to block the effects of an inhibitory factor for wild-type stable Arm to translocate to the nucleus. ArmS10 is missing the interacting region and thus does not require the protective function of the Kinesin-2/IFT-A complex in nuclear translocation, and ArmS10 can reach the nucleus passively or by other means (see Discussion).

**IFT140 function on β-cat in Wnt-activated cells is conserved.** To determine whether the above-proposed IFT-A-dependent mechanism for nuclear translocation of β-catenin is conserved in mammals, we tested the respective factors in wild-type and mutant mouse embryonic fibroblasts (MEFs). We asked whether nuclear translocation of β-catenin depends on IFT140 in MEFs. Wild-type (*WT*) MEFs revealed detectable levels of nuclear β-catenin upon Wnt3a treatment (Fig. 8a, b), similarly to the Wg-induced translocation of Arm/β-catenin in *Drosophila* salivary glands (Fig. 5). However, under the same Wnt3a treatment, *ift140^−/−* mutant MEFs showed a striking reduction of nuclear β-catenin (Fig. 8c, d, quantified in Fig. 8i; note loss of IFT140 protein in Supplementary Figure 7e), indicating that the IFT140 function for nuclear β-catenin translocation is conserved. Consistently, this IFT140 requirement was Wnt3a dependent, as without Wnt3a-treatment neither cells displayed nuclear β-catenin (Fig. 8b, d). While the effect did not depend on presence or absence of cilia or cellular density of the MEFs (Supplementary Figure 7a–d), we wished to address its specificity for IFT-A complexes vs. IFT-B, by comparing it to inducible *ift88* knockout MEFs. IFT88 is a component of IFT-B, which does not affect Wnt/Wg-signaling in *Drosophila* but does affect cilia formation[20]. *ift88* knockout (via *CreER^T2* with Tamoxifen; and vehicle-only control) had no effect on nuclear β-catenin localization following Wnt3a treatment (Fig. 8e−h; quantified in Fig. 8i), reflecting normal nuclear β-catenin levels, with or without *ift88* presence (Supplementary Figure 7f shows efficient tamoxifen-mediated IFT88 loss). These data indicate a conserved and specific IFT-A (*ift140*) requirement for β-catenin nuclear translocation, and further demonstrates that presence of cilia is not required for Wnt-signaling and nuclear β-catenin localization, as *ift88^−/−* MEFs completely lack cilia (Supplementary Figure 7d), and yet Wnt-signaling and nuclear β-catenin is unaffected (Fig. 8g).

To confirm this, we analyzed nuclear β-catenin levels by western blot in these backgrounds. Whereas wild-type and uninduced *CreER^T2ift88^−/fx* control MEFs displayed strong accumulation of nuclear β-catenin upon Wnt3a treatment, *ift140^−/−* MEFs displayed markedly reduced nuclear β-catenin levels (Fig. 8j; note unaffected total β-catenin levels in all backgrounds).

To determine whether mammalian β-catenin displays a conserved behavior to the *Drosophila* ArmS10 deletion in mammalian Wnt-signaling, we generated β-catFL (full length), β-catS10 and β-cat* (both equivalent to the respective *Drosophila* mutations, see Fig. 6k, tagged with GFP) and analyzed their

nuclear β-catenin levels in the respective MEF backgrounds by western blot. β-catS33Y-GFP served as a control for a known "activated" β-catenin mammalian mutant[45], since it was suggested to have the same "function" as ArmS10 in *Drosophila*.

Wild-type MEFs as well as induced *CreER^T2ift88^−/fx* control MEFs showed accumulation of nuclear β-catenin in cells transfected with the stable β-catS10-GFP, β-cat*-GFP, and β-catS33Y-GFP (Fig. 9a, c, d). As a control, full-length (wild-type) β-catenin (β-catFL-GFP) was detected in the cytoplasm and absent from the nuclear fraction (Fig. 9a−d; left most lanes in all panels), indicating that β-catS10-GFP, β-cat*-GFP, and β-catS33Y-GFP were stable β-cat isoforms, translocating readily into the nucleus. In contrast, in *ift140^−/−* MEFs only β-catS10-GFP displayed nuclear β-catenin accumulation, whereas β-cat* and β-catS33Y, like wild-type β-catFL remained cytoplasmic (Fig. 9b). These data indicated that β-catS10 behaves indistinguishably from *Drosophila* ArmS10, not requiring the IFT-A-mediated process for nuclear translocation.

These data support three conclusions: (i) the requirement of the Kinesin-2/IFT-A complex for nuclear translocation of β-catenin is conserved between *Drosophila* and mammals, (ii) this function is specific to IFT-A and not observed with the IFT-B complex, and (iii) the requirement of amino acids deleted in the ArmS10/β-catS10 isoform for cytoplasmic retention is also conserved. In summary, these data support a non-ciliary mechanistic function of IFT-A complex components in Wnt-signaling that is evolutionarily conserved.

## Discussion
The IFT-A complex is best known for its function in ciliary transport[46,47], and it has been suggested to promote canonical Wnt/Wg-signaling independently of its ciliary role in *Drosophila*[20]. In the latter context it remained unclear whether (i) IFT-A also associates with microtubule structures via kinesin in the cytoplasm, (ii) how this complex mechanistically promotes Wg/Wnt–signaling, and importantly (iii) whether this IFT-A function is evolutionarily conserved in mammals. A function of Klp64D, the *Drosophila* Kif3A Kinesin-2 subunit, was also reported in the context of Wnt-signaling and Arm/β-Catenin trafficking in *Drosophila*[19]. This raised the possibility that IFT-A may interact with microtubules via Kinesin-2 and share a function in Wg/Wnt–signaling, possibly similar to Costal-2 (Cos2)/Kif7 in Hedgehog signaling, where Cos2 sequesters Ci/Gli proteins to microtubules[48,49].

Here we demonstrate the association between IFT-A, Kinesin-2, and Arm/β-Catenin and define a conserved mechanistic role of this complex in Wnt-signaling: the Kinesin-2/IFT-A complex is required to promote nuclear translocation of Arm/β-Catenin both in *Drosophila* and mammals to achieve high signaling levels. Note that while *Sens* expression is lost, low levels of signaling appear independent of this mechanism, as *Dll*, for example, is still expressed albeit at reduced levels. Kinesin-2 and IFT-A associate physically and act synergistically in vivo, and importantly the

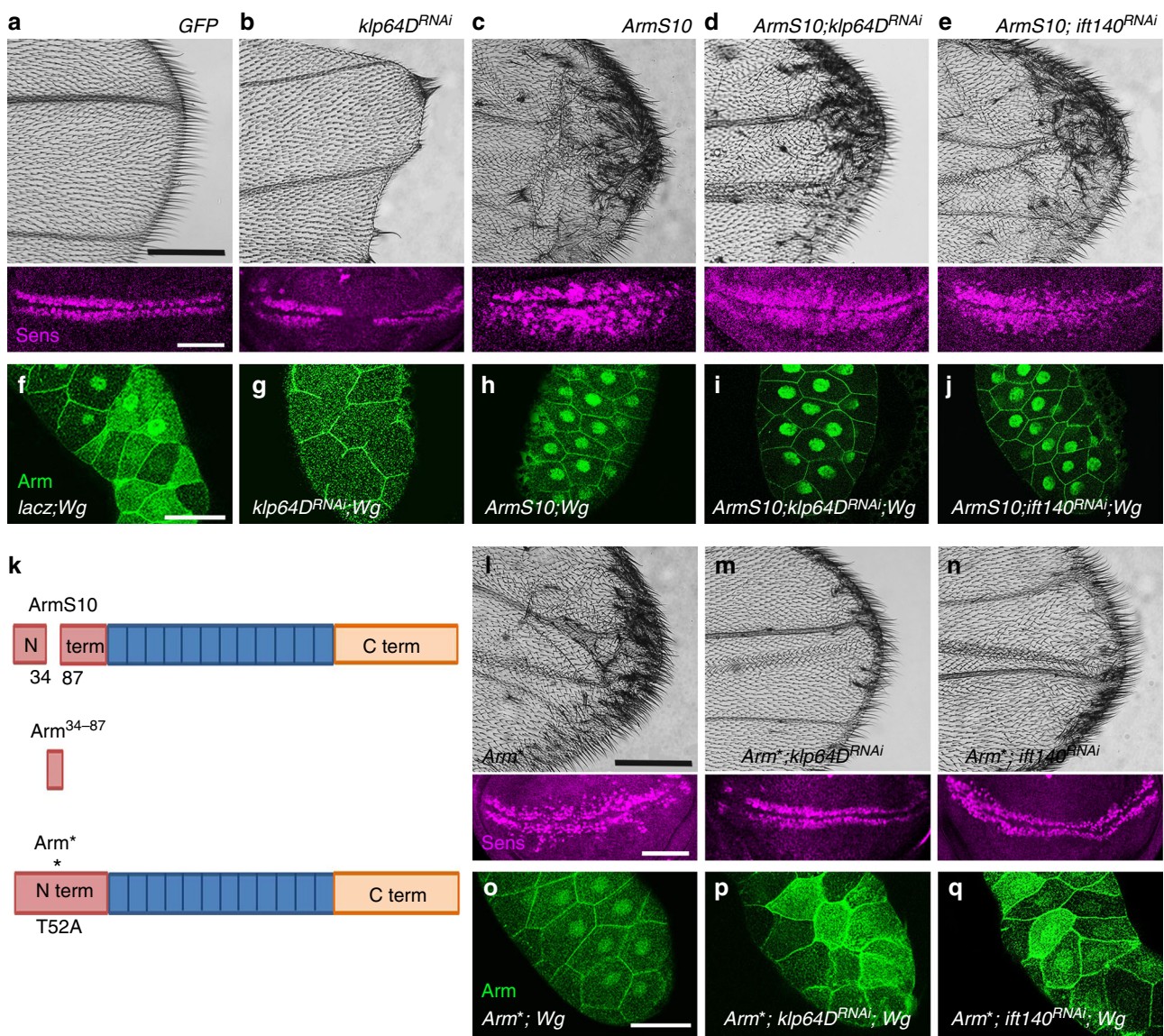

**Fig. 6** Stable Arm vs. ArmS10 behavior with respect to Kinesin-2/IFT-A complex function. Panels (**a–e**) and (**l–n**) show distal regions of adult wings (top) and D/V margin in wing discs of same genotypes (bottom). Anti-Sens (magenta in bottom panels) marks future margin bristles. **f–j** and **o–q** show salivary glands with *C805-Gal4, UAS-Wg*; anti-Arm in green (**f–j** and **o–q** serves also as cellular outline due to jucntional staining. **a** *C96 > GFP* wing (control), with wild-type wing margin and Sens expression in imaginal discs. **b** *C96 > klp64D^RNAi*: wing notching phenotype and reduced Sens expression. **c** *C96 > ArmS10*: ectopic wing margin bristle phenotype and ectopic Sens expression in discs. **d** *C96 > ArmS10, >klp64D^RNAi*: largely no effect both in adult wings and imaginal discs. **e** *C96 > ArmS10, >ift140^RNAi*: note ectopic wing margin bristles and Sens expression, very similar to ArmS10 expression alone. **f–j** Arm levels/ArmS10 behavior and localization of the respective genotypes in the salivary gland Wg-signaling assay. **f** *C805-Gal4 > lacz, > Wg* salivary gland positive control: note increased Arm/β-Catenin levels and localization in both cytoplasm and nucleus. **g** Nuclear Arm/β-Catenin localization is reduced when knocking down Klp64D. **h** Salivary glands display Arm/β-Catenin (ArmS10) mainly in nucleus. **i** Knockdown of *Klp64D* in ArmS10 background salivary glands, with ArmS10 not being affected and localized mainly to the nucleus. **j** Knockdown of *IFT140* in ArmS10 salivary glands, note ArmS10 is not affected and localized to the nucleus. **k** Domain structure of Arm/β-Catenin, highlighting the deletion in ArmS10 (top), small Arm[34-87] fragment (middle), and point mutation in Arm*. **l** *C96 > Arm** (Arm* is a stable, active Arm/β-Catenin with point mutation in the destruction complex target sites); note ectopic wing margin bristles, and ectopic Sens expression. **m** Knockdown *Klp64D* in the *C96 > Arm** background suppresses the ectopic margin bristle phenotype and Sens expression. **n** Knockdown of *ift140* in *C96 > Arm**: note suppressed ectopic margin bristles and Sens expression. **o** Arm* in *wt* salivary glands is localized mainly to the nucleus. **p** Knockdown *Klp64D* in Arm* background salivary glands causes markedly reduced nuclear Arm localization. **q** Knockdown *IFT140* in Arm* salivary glands: note markedly reduced nuclear Arm/β-Catenin localization. Scale bars represent 100 μm (**a–n**; top), 30 μm (**a–n**; bottom) and 50 μm (**f–q**). All the tested wings show phenotype (n = 100)

process requires the motor function of Klp64D, suggesting that movement along microtubules is essential. This is further supported by subcellular colocalization of IFT-A and Arm/β-Catenin punctae, which decorate filamentous structures, like "beads on a string" (Fig. 4b). The nonmotor Kap3 subunit of Kinesin-2 functions as an adapter that directly binds the IFT-A complex, a

role that is similar to IFT-A/Kinesin associations in ciliary transport, where IFT-A serves as a cargo loading platform/ adapter for kinesin. The motor activity and microtubule association of these complexes appear critical for Wg/Wnt-signaling (this work and ref. [19]). Our data thus suggest that a Kinesin-2/ IFT-A complex is required to transport Arm/β-catenin,

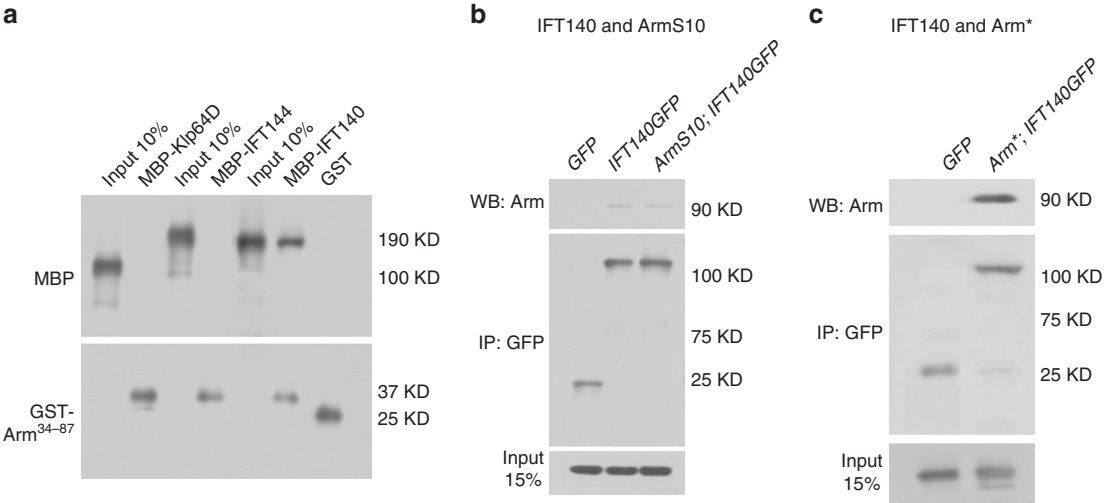

**Fig. 7** Physical interaction between Klp64D, IFT140, and Arm34-87. **a** GST pull-down between Klp64D or IFT140 and Arm34-87. IFT140 directly binds to amino acids 34−87 of Arm, whereas Klp64D does not. IFT144 does not bind to Arm34-87, as a negative control. **b**, **c**) Co-immunoprecipitation assay of IFT140 and ArmS10 from wing imaginal discs of *C96 > GFP*, *C96 > ArmS10; > IFT140-GFP* (both in **b**), and *C96 > Arm\**; *>IFT140-GFP* (panel (**c**)) genotypes. Protein extracts from wing discs were IP-ed with anti-GFP. The immune complexes and input (15% of wing disc lysates used in IP step) were analyzed by immunoblotting with anti-Arm antibody. **b** IFT140 does not bind ArmS10; note that the weak background band is also present in *wt* control GFP lane and represents endogenous Arm (middle lane; as confirmed by its migration behavior; ArmS10 is a deletion and migrates faster). **c** IFT140 forms strong association with Arm\*

independently of the cilium, and this function is conserved in mammals and promotes nuclear translocation of Arm/β-Catenin.

Wg/Wnt pathway activation induces relocation of Axin from the "destruction complex" to the plasma membrane, resulting in the release of Arm/β-catenin and its stabilization in the cytoplasm. As a consequence, Arm/β-Catenin enters the nucleus and stimulates transcription of Wnt/Wg target genes[1–4]. Although this pathway is highly conserved and fairly well dissected, how Arm/β-catenin is translocated to and retained in the nucleus remain poorly understood. It has been suggested that β-catenin is imported via a nuclear localization signal by interacting directly with nuclear pore proteins[50]. It has also been proposed that the β-catenin nuclear and cytoplasmic states are regulated by stimulation of both processes, Axin and APC retain β-catenin in the cytoplasm and TCF and BCL9 (a β-catenin co-activator) maintain β-catenin levels in the nucleus via multiple mechanisms[51]. It was further proposed that Rac1, JNK2, and β-catenin form a complex in the cytoplasm and JNK2 phosphorylates β-catenin to promote its nuclear translocation[52], although it remains unclear how Rac1, normally linked to Wnt/Fz-PCP signaling[53,54], is involved in canonical Wnt-signaling. We demonstrate a requirement of Kinesin-2/IFT-A for nuclear translocation of Arm/β-catenin. Strikingly, double mutant clones for either Kinesin-2 or IFT-A components with *axin* displayed high levels of stabilized non-junctional Arm, same as in *axin* single mutant clones, but signaling (as detected in wing discs) and Arm/β-catenin nuclear translocation was markedly reduced, both in salivary gland Wg-signaling assay and mouse cells (MEFs). Given the function of Kinesin-2 as a motor protein, this suggested that the Kinesin-2/IFT-A complex promotes microtubule-based transport of Arm/β-catenin towards the nucleus. This hypothesis was corroborated with motor domain Kinesin-2 mutants, which displayed the same Wnt-signaling defects as Kinesin-2 or IFT-A null alleles. Importantly, this IFT-A function is conserved and independent of cilia.

Physical interaction studies indicate that the Kinesin-2/IFT-A complex directly interacts with Arm/β-catenin through IFT140, with IFT-A acting as a cargo adapter for Kinesin-2. The ArmS10 isoform, the stable "constitutively active" isoform of Arm/β-Catenin, is a small deletion within the N-terminal region, which is phosphorylated by the destruction complex to target it for degradation. Strikingly, ArmS10 behaves differently from other stable Arm/β-catenin proteins, which carry point mutations that block CK1 phosphorylation of Arm/β-catenin, but are otherwise full-length (ref. [44]). ArmS10 can enter the nucleus independently of the Kinesin-2/IFT-A function and, moreover, does not bind to the complex. In fact, the deleted 53 amino acid region of ArmS10 is sufficient to interact with IFT140 (IFT-A) on its own. This suggests that the small deletion in ArmS10 is critical for its differential behavior relative to *wt* Arm/β-catenin in terms of Kinesin-2/IFT-A requirements. The behavior of the equivalent mammalian β-catenin stable mutant isoforms is identical to *Drosophila* Arm/ArmS10, again supporting a fully conserved mechanism.

As binding of Arm/β-catenin to the Kinesin-2/IFT-A complex promotes nuclear localization, it is likely that IFT140/IFT-A competes with another protein for Arm/β-catenin binding in this specific domain, and that this cytoplasmic protein could trap/sequester Arm/β-catenin in the cytoplasm (and reduce its capacity to enter the nucleus). ArmS10 and mammalian β-cateninS10 equivalent forms are indifferent to this mechanism and can translocate to the nucleus passively or through another mechanism, because the binding region for both factors, positive (IFT140) and antagonistic unknown protein, is eliminated.

These observations raise two intriguing questions that will be the focus of future studies: (i) can the deleted residues of ArmS10/ β-cateninS10 be developed as a therapeutic target to trap hyperactive β-catenin in the cytoplasm, and (ii) identification of the endogenous factor(s) that bind this region of Arm/β-catenin to inhibit nuclear translocation. Axin was suggested as a cytoplasmic anchor keeping Arm/β-catenin out of the nucleus[55] and our data in the Wg-signaling salivary gland assay are consistent with this, as almost all Arm/β-catenin is found in the nucleus in *axn*[RNAi] background, whereas more is retained in the cytoplasm upon pathway activation via Wg-expression (cf. Fig. 5a, f). However, Axin can be excluded for this function, because in the absence of Axin stable Arm/β-catenin still fails to translocate to the nucleus

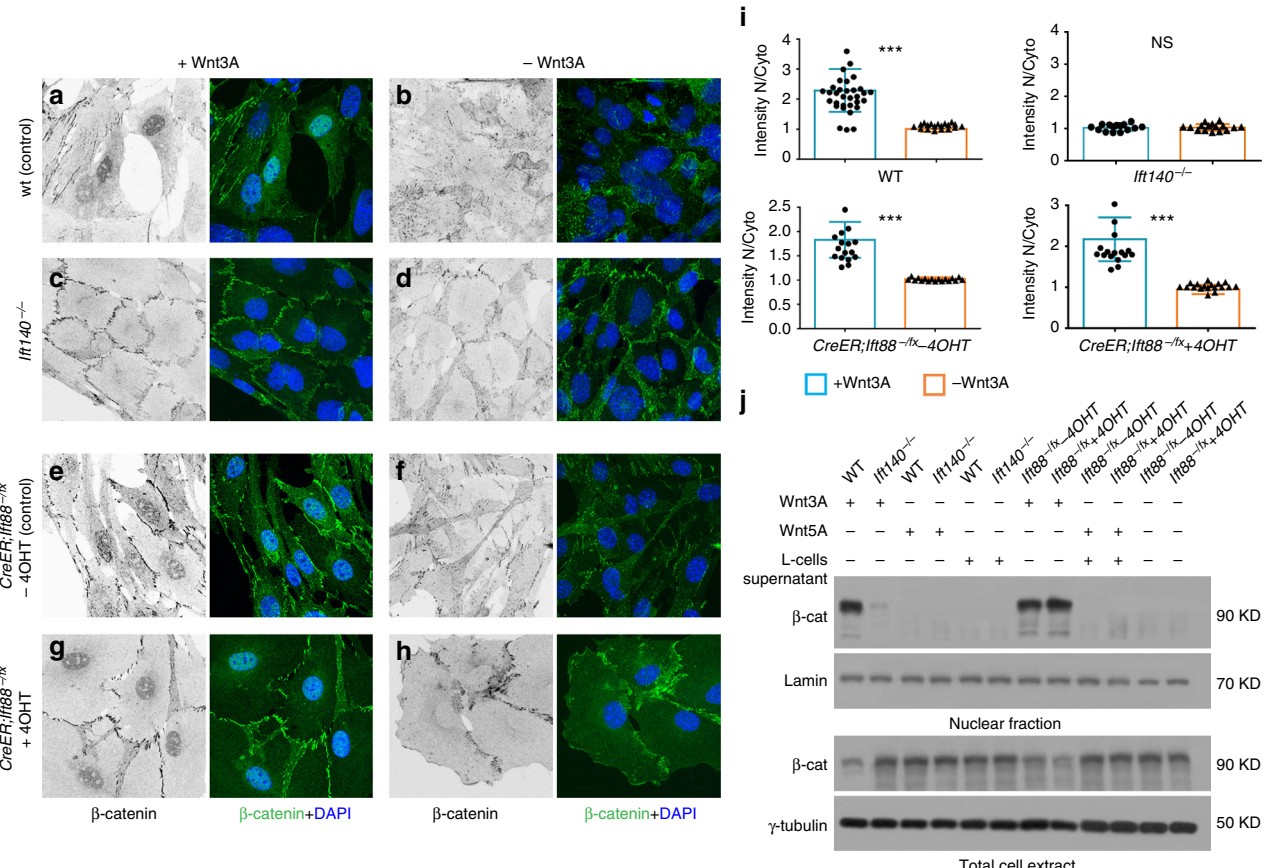

**Fig. 8** IFT140, but not IFT88 (IFT-B complex), is required for Wnt3A-induced nuclear accumulation of β-catenin in MEFs. **a−h** Confocal images of immunofluorescence (IF) of β-catenin (monochrome on left, green in overlays, right) and nuclei (blue in overlay) in MEFs treated with Wnt3a-conditioned (**a**, **c**, **e**, **g**) or control conditioned media (**b**, **d**, **f**, **h**) media. Upon 24 h of Wnt3A stimulation, β-catenin becomes readily detectable in nuclei of wild-type MEFs (**a**) and tamoxifen inducible *CreER; Ift88*[−/fx] MEFs cultured in absence (**e**) or presence (**g**) of 50 nM 4-hydroxytamoxifen (4OHT); in contrast, it remained undetectable in nuclei of *Ift140*[-/-] MEFs (**c**). No nuclear β-catenin was detected in any MEFs cultured with control media (**b**, **d**, **f**, **h**). IF signal of membrane-associated β-catenin was undistinguishable in all MEFs, cultured with or without Wnt3A-conditioned media (**a−h**). None of the MEFs were treated with LepB. Scale bar = 80 μm for all panels (**a−h**). **i** Quantification of nuclear β-catenin IF signal in MEFs. *Y* axis denominates the ratio of gray intensity values of selected regions (area of 4.705 μm × 4.843 μm) within nuclei and cytoplasm of individual cells (membrane-associated β-catenin was purposely excluded). Mean + s.d. of values obtained in randomly selected cells are shown from three independent experiments; Student's *t* test: *** indicate $p < 0.001$; n.s.: difference not statistically significant. **j** Wnt3A-induced nuclear accumulation of β-catenin as determined by western blots of nuclear and cytoplasmic cell fractions of control and mutant MEFs grown in Wnt3A-conditioned or control media (Wnt5A or L-cells supernatant). IFT88 was induced by *CreER* driver. γ-tubulin and Lamin B were used as loading controls. Note marked reduction of nuclear of β-catenin in *ift140*[−/−] MEFs. Wnt5a condition media (Wnt5a does not induce canonical Wnt-signaling in these cells) was used as negative control. Also note that in lanes where nuclear localization of β-catenin is detected, the levels in the cytoplasm are reduced

in *Klp64D* or *ift140* mutants. Efforts to identify an IFT140 competing factor(s) await further investigation.

The Wnt canonical pathway is commonly activated by constitutive upregulation of wild-type or phosphorylation mutants of β-catenin that escape degradation via the destruction complex. With this study we have identified a process in the path that Arm/β-catenin requires on its way to the nucleus and one that will be of interest to biologists and clinical researchers alike. Our discovery that the IFT-A/Kinesin-2 complex plays a critical role in canonical Wnt-signaling via Arm/β-catenin identifies possible therapeutic targets for a wide spectrum of human malignancies, in which a hyperactive β-catenin is an oncogenic driver.

## Methods

**Fly stocks**. The *GAL4 /UAS* system was used for expression of RNAi and constructs, or in combination *UAS-Dcr2* (where indicated). GAL4 driver for wing margin during wing development was C96-GAL4 (BL43343) or *UAS-Dcr2; C96-Gal4* (BL25757) expressed around dorsal-ventral boundary of wing imaginal discs.

Other *GAL4* lines used were: *tub-Gal4* and salivary gland-specific Gal4 line (PI)(2) C805[C805] (BL6986) or (Pc135) (BL6978). All crosses were set up at 29 °C, unless otherwise indicated.

The following transgenic lines were used:
*UAS-IFT121GFP* and *UAS-IFT140GFP* or *oseg1*[179] (gifts from T. Avidor-Reiss)
*UAS-IFT121myc* and *UAS-IFT140myc* (from S. Balmer paper)
*UAS-Klp64DHA* and *UAS-Klp64D*[ΔABD] or *UAS-Klp64D*[T114A] (gifts from K.W. Choi)
*UAS-Arm*\* (gift from K. Cadigan)
*rempA*[l(2)21Ci-1] (gift from M. Kernan)
*tub-exd*[V5] (gift from R. Mann)
*UAS-Klp64D* (BL32008)
*UAS-Klp64D RNAi* (BL40945 and v103358)
*UAS-Kap3 RNAi* (v4540 and v103548)
*UAS-IFT121 RNAi* (*oseg4*) (v1098905)
*UAS-IFT122RNAi* (*oseg1*) (v103598)
*UAS-IFT140 RNAi* (*rempA*) (v31575)
*UAS-IFT 143 RNAi* (v106366)
*UAS-Axin RNAi* (BL31705)
*UAS-Axin GFP* (BL7224)
*UAS-ArmS10* (BL4782)
*axin*[e77] (BL17649)

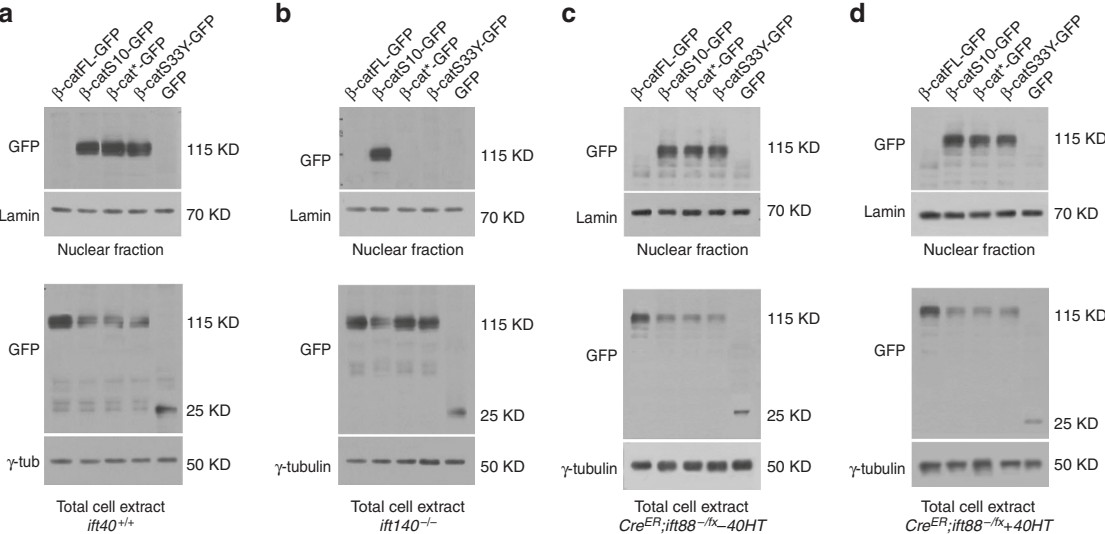

**Fig. 9** β-cat* and β-catS33Y stable isoforms require IFT140 for nuclear localization in MEFs. **a–d** Nuclear and/or cytoplasmic accumulation of full-length β-catenin (β-catFL-GFP), and mutant isoforms of β-cat tagged with GFP (β-catS10-GFP, β-cat*-GFP, and β-catS33Y-GFP) or GFP alone were determined by western blots of nuclear and cytoplasmic cell fractions (without treatment of Wnt3A). γ-tubulin and Lamin B were used as loading controls. Note that the β-catS10 activated isoform localizes to the nucleus independent of the ift140 activity, whereas the β-cat* and β-catS33 isoforms require IFT140 (compare panels (**a**) and (**d**)). None of the activated β-catenin isoforms requires IFT88 (compare panels (**c**) and (**d**))

klp64D[k1] (BL5578)
y w, hsflp; ubiGFP FRT2A, FRT82B/TM6b (BL5936)

**Antibodies and staining reagents**. For tissue staining, primary antibodies used were:

guinea pig anti-Sens (1:500, gift from H. Bellen); rat anti-Dll (1:250 from Jun Wu); rabbit anti-IFT140 (1:500, gift from G. Pazour); mouse anti-Arm (1:10, DSHB 7A1); rat anti-DE-cad (1:50, DSHB DCAD2); rat anti-Dll (1:250); mouse anti-myc (1:200, Santa Cruz, 9E10); rabbit anti-myc (1:200, Santa Cruz, d1-717); chicken anti-GFP (1:200, Aves Labs 1020); rabbit anti-RFP (1:1000, Rockland); rabbit anti-Dlg (1:200, DSHB); mouse anti-HA (1:50, Roche); rabbit-anti-β-catenin (1:100, Invitrogen AHO0462); rabbit-anti-IFT88 (1:1000, Proteintech Group 13967-1-AP); rabbit anti-Arlb13 (1:800, Proteintech Group 17711-1-AP); goat anti-γ-tubulin (1:200, Santa Cruz sc-17787) and Hoechst 33342 (10 mg/ml, 1:500).

For immunoblotting, the following primary antibodies were used:

mouse anti-Axin (1:500) (gift from Nusse); rabbit anti-Klp64D (1:500) (gift from Kwang Choi); mouse anti-Arm (1:1000, DSHB 7A1); mouse anti-γ-tubulin (1:1000, Sigma); mouse anti-GFP (1:1000, Roche); rabbit anti-IFT88 (1:2000, Proteintech Group); mouse anti-GST (1:1000, Invitrogen); and rabbit anti-MBP (1:1000, Sigma); rabbit anti-IFT140 (1:1000, Proteintech Group); rabbit anti β-catenin (1:800, Invitrogen); rabbit anti-laminb1 (1:1000, Abcam); rabbit anti-Axin2 (1:1000, ProSci).

All fluorescent secondary antibodies and HRP-coupled secondary antibodies were from Jackson Immuno Research Laboratories and used at 1:200.

**Immunostaining and histology**. Imaginal wing discs were dissected at third instar larval stage in phosphate buffered saline (PBS) and fixed in PBS, 4% paraformaldehyde (PFA). Discs were washed two times in PBS 0.1% Triton-X100 (PBT), incubated in primary antibodies overnight at 4 °C. After washing in PBT, incubation with secondary antibodies was at room temperature for 2 h. Samples were mounted in Vectashield (Vector Laboratories). Wing disc images were acquired with a confocal microscope (×20–40, oil immersion, Leica SP5IDM or Zeiss LSM880 system). Images were processed with ImageJ (National Institutes of Health) and Photoshop (CS4; Adobe).

Salivary glands were dissected at the third instar larval stage in PBS and treated with 0.1% Leptomycin B (Sigma) in 10 min before fixation in PBS, 4% PFA. All subsequent steps were as described for wing imaginal discs. Colocalizations were performed using step size of 0.5× relative to the optical section thickness, according to the Nyquist theorem sampling criterion[56], as implemented by the Microscopy Core facility.

*Analyses of adult wings*: wings were removed, incubated in PBS with Triton detergent, and mounted on a slide in 80% glycerol in PBS, and imaged using Zeiss Axioplan microscope. All adult images were acquired using Zeiss Axiocam color-type 412-312 camera and the Zeiss axiocam software.

**GST-pull-down assay**. For GST pull-downs, IPTG-inducible *E. coli R2* cells (*BL21* derivative) were transformed with plasmid constructs for fusion proteins MBP-Axin, MBP-Klp64D, MBP-Klp61F, MBP-Kap3 (Kap3cDNA (LD13052)), GST-

IFT140, GST-Kap3, GST-Axin, and GST- Klp64D. Bacterial lysates were prepared as described in ref. [56]. Equal amounts of blocked glutathione Sepharose 4B beads (Bioprogen) with GST, GST fusion protein, or beads alone were incubated in pull-down buffer (20 mM Tris pH 7.5, 150 mM NaCl, 0.5 M ethylenediaminetetraacetic acid (EDTA), 10% glycerol, 0.1% Triton X100, 1 mM dithiothreitol (DTT) and protease inhibitor cocktail), 1× sample buffer was added, beads were boiled, and proteins were resolved on sodium dodecyl sulfate polyacrylamide gel electrophoresis (SDS-PAGE). Proteins were electrophoretically transferred onto nitrocellulose, blocked in 5% skim milk (Bio-Rad) for 1 h, and incubated with primary mouse anti-GST (Santa Cruz) or rabbit anti-MBP antibody (Santa Cruz).

**Immunoprecipitation**. Lysates from 120 wing imaginal discs of *tub > AxinGFP*, *tub > IFT122GFP*, *tub > IFT140GFP* (1 mg of total protein) were precleared by incubating with protein A-sepharose beads (Thermo Scientific) for 1 h at 4 °C followed by centrifugation. A-sepharose beads were immunoprecipitated with specific antibodies at 4 °C for 1 h. Polyclonal anti-GFP antibody (Roche) was used. Immunoprecipitates were captured by protein A-sepharose at 4 °C in IP buffer (20 mM 4-(2-hydroxyethyl)-1-piperazineethanesulfonic acid (HEPES) pH 7.5, 100 mM NaCl, 0.05% Triton X100, 1 mM EDTA, 5 mM DTT, 10 mM NaVO4, 10% glycerol and protease inhibitor cocktail). Immunoprecipitates were resuspended in SDS sample buffer, boiled for 5 min, separated by SDS-PAGE, and transferred to nitrocellulose for immunoblotting. Protein was detected by enhanced chemiluminescence (Millipore).

**MEF isolation and Wnt3a-induced β-catenin nuclear accumulation**. MEFs carrying a tamoxifen-inducible knockout of *Ift88* were obtained from mice containing the loxP and null *Ift88* alleles[57] crossed to the CAGG-creERTM mouse[58] and maintained on a C57BL/6 background. MEFs were collected from E13.5 mouse embryos killed by cervical dislocation and dissected into cold Dulbecco's modified Eagle's medium (DMEM). Embryonic tissue not including the head and visceral organs was incubated in 0.25% trypsin for 10–15 min at 37 °C. Dissociated MEFs were grown in DMEM medium supplemented with 10% fetal bovine serum at 37 °C in a 5% CO2 incubator. Cells at 80–85% confluence were trypsinized, washed with PBS and plated for each experiment. To delete *Ift88* in *CreER; Ift88[−/fx]* MEFs, 50 nM 4-hydroxytamoxifen (4OHT) solubilized in ethanol was added to the medium for 24 h. Ethanol-treated *CreER; Ift88[−/fx]* MEFs were used as control. *Ift140[−/−]* MEFs were a gift from Dr. Gregory Pazour (University of Massachusetts).

For Wnt3A-induced β-catenin nuclear translocation, MEFs were grown at 70% confluence in DMEM medium supplemented with 10% fetal bovine serum and treated with Wnt3A-conditioned medium for 14–16 h as described in ref. [59]. Cells were then fixed in cold methanol for 15 min at −20 °C and labeled with primary antibodies for β-catenin diluted 1:100 in 2% bovine serum albumin/PBS for 2 h, washed in PBS, incubated with fluorescent secondary antibodies (Alexa Fluor 488, Life Technologies, Carlsbad, CA) diluted 1:500 in PBS for 1 h and mounted with VECTASHIELD® mounting medium containing 4′,6-diamidino-2-phenylindole.

**Quantification of β-catenin nuclear translocation.** Confocal imaging was performed using a Zeiss LSM 880 Airyscan mounted on a Zeiss Axio-Observer inverted microscope. Z-stacks of 6 (1.083 μm) to 12 optical slices (4.362 μm) at 8-bit were captured and analyzed via ImageJ-Fiji. β-catenin translocation was evaluated through optical density assays of mean gray values between the nucleus and cytoplasm of individual cells. To maintain uniformity in all acquisitions, measurements were obtained under an image size of 1024 × 1024 pixels, FITC (green) channel, maximum projection (Z-project on Image J), and a standardized region of interest at 4.705 μm × 4.843 μm avoiding cell membrane regions. Intensity ratios (*nucleus/cytoplasm*), standard deviations, and Student's two-tailed test were performed using Microsoft® Excel.

**β-catenin nuclear fraction assay.** Wnt3A stimulated or unstimulated MEFs were gently washed with PBS buffer and cells were minced on ice by sharp scalpel and collected by centrifugation at 5000 rpm/4 °C. All samples were resuspended in 500 μl buffer comprising 250 mM sucrose, 50 mM Tris-Cl pH 7.4, 5 mM $MgCl_2$, 1 M EDTA, 1% TritonX100 and protease inhibitor cocktail (Roche) and gently homogenized for 1 min on ice using homogenizer. Samples were transferred to microfuge tube and kept on ice for 30 min at 4 °C. Supernatants were cleared by 20 min centrifugation at 4 °C and saved as the cytoplasmic fraction. The pellet or nuclei fraction was washed again by the lysis buffer above without protease inhibitor cocktail. Nuclear fraction was incubated for 30 min at 4 °C with nuclear extract buffer containing 20 mM HEPES pH 7.9, 15 mM $MgCl_2$, 0.5 M NaCl, 1 M EDTA, 20% glycerol, 1% TritonX100, protease inhibitor cocktail and sonicated for 10 min after incubation. The supernatant was collected by 30 min centrifugation at 4 °C as the nuclear fraction. Protein extracts were boiled for 5 min at 95 °C in SDS-sample buffer, separated by 10% SDS-PAGE gel and transferred to nitrocellulose membrane. Protein levels were analyzed by immunoblotting with the corresponding antibodies.

For the β-catenin mutant isoform nuclear fraction experiments: The β-cateninFL-GFP, β-cateninS10-GFP, β-catenin*-GFP and β-cateninS33Y-GFP constructs were made from the original plasmid MSCV- β-catenin-IRES-GFP (Addgene Plasmid #14717) and pMXs- β-catenin-S33Y (Addgene Plasmid # 1371). All constructs were transfected into MEFs by Lipofectamine LTX & Plus™ Reagent (Invitrogen A12621). Cells were collected 12 h after transfection. The nuclear fraction assay was performed as mentioned above.

All the original western blot gels are provided in Supplementary Fig. 8.

## Data availability

The data that support the findings of this study are available from the corresponding authors upon reasonable request.

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

## Acknowledgements

We thank the Bloomington Stock Center, Vienna Drosophila RNAi center, Drosophila Genomics Resource Center (DGRC) and Developmental Studies Hybridoma Bank for fly strains and antibodies. We are most grateful to Ken Cadigan for helpful discussions and *UAS-Arm\** flies; Bradley Yoder and the University of Alabama/Hepatorenal Fibrocystic Disease Core Center (P30 DK074038) for providing *the Ift88*$^{tm1Bky}$ (here referred to as *Ift88*$^{fx/fx}$) and the null *Ift88* mouse alleles; and K.W. Choi for anti-Klp64D antibody and *UAS-Klp64D*$^{\Delta ABD}$, *UAS-Klp64D*$^{T114A}$, and *UAS-Klp64D-HA* fly strains and Kinesin-2 and Arm constructs for GST pull-down assays. We thank H. Bellen for anti-Sens antibody; L. Gusella for anti-IFT88; G. Pazour for providing *Ift140*$^{-/-}$ MEFs and anti-IFT140 antibody; R. Nusse for anti-Axin; T. Avidor-Reiss for *IFT121-GFP*, *IFT140-GFP* and *oseg1*$^{179}$ flies; M. Kernan for *rempA*$^{21Ci}$; and Richard Mann for *tub-exdV5* flies, respectively. We thank Jong-Sun Kang, Qing Liu, Varun Arvind, and Anna Yang for their help in experiments involving MEFs, and all Mlodzik and Iomini lab members for helpful inputs and discussion, and A. Humphries, U. Weber, and G. Collu for critical comments on the manuscript. Confocal microscopy was performed at the Tisch Cancer Institute Microscopy Core, supported by grant P30 CA196521 from the NCI. This work was supported by National Institutes of Health grants GM127103 to M.M., EY022639 to C.I., HD066319 to M.M. and S.A.A., and a Breast Cancer Research Foundation grant to S.A.A.

## Author contributions

L.T.V., C.I., and M.M. conceived and designed the study, also based on discussions with S.B. L.T.V. and C.I. performed the experiments. S.B., D.E., and S.A.A. provided critical reagents and input to Drosophila and MEF experiments, respectively. C.I., S.A.A. and M.M. provided funding for the project. L.T.V., C.I., and M.M. analyzed and interpreted the data, and wrote the manuscript.

## Additional information

**Competing interests:** The authors declare no competing interests.

