## [Peer Review File · Nature Communications]

Reviewers' Comments:

Reviewer #1:

Remarks to the Author:

The paper by Vuong et al proposes that a kinesin-2/IFT complex can stimulate wnt signaling through direct binding to beta-catenin and by promoting its translocation to the nucleus through a microtubule motor dependent function - and that this occurs in drosophila and mouse embryo fibroblasts. I found certain sections of this paper to be excellent, with a very thorough genetic dissection of the interactions between the kinesin-2 and IFT-A complex components. They present reasonable evidence that kinesin-2 (ie. klp64d) and the ciliary factor IFT-140 can influence nuclear levels of beta-catenin under specific circumstances following wnt stimulus. However there is no evidence in this paper to show that klp64d or IFT140 affect the rate of nuclear transport. The authors performed no actual cell assays to analyse nuclear import or export, or even standard CSK extraction assays to test for changes in nuclear or cytoplasmic retention of beta-catenin. The main conclusion to be drawn from the limited data shown is that there is some effect on nuclear beta-catenin, most likely due to altered retention in the cytoplasm. This is interesting, but without a mechanism it truly lacks impact.

Major comments about Figures:

1. Figures 1 and 2 look okay.
2. Figure 3f model. This diagram is partly supported by the data but there are some issues. (a) include an MBP-only control in Figure 3B, as beta-catenin is a promiscuous protein and it is important to exclude non-specific interactions. Other MBP-fusions do not directly address this issue. (b) The sentence in lines 205-207 suggest there is data showing that KAP3 and klp64d do not bind to beta-catenin in the Supp Fig 3, however I could see no such data presented. (c) In addition, this statement contradicts the finding previously published by the first author, L. Vuong in Ref 19, showing that klp64D binds directly to beta-catenin via its Armadillo repeat domain.
3. Figure 4. Co-staining of ectopic klp64d-HA and IFT140-Myc. The cell images look overexposed, obviously to boost signal. But the co-staining looks sporadic and such images need some quantification to demonstrate the extent of co-localization in a larger population of cells (at least 50-100). Also in Supp Fig 4d, there are blobs of IFT140/Arm co-staining that look less like specific "puncta" and more like artificial aggregates caused by protein overexpression.
4. Figure 5. The authors used larger salivary glands to detect nuclear staining, however they employed use of the CRM1 inhibitor leptomycin B (normally abbreviated LMB) to boost the visible nuclear beta-catenin levels after wnt stimulation. As shown in the previous (but mostly uncited) literature, beta-catenin is a shuttling protein and its nuclear localization can be increased indirectly by LMB through forced retention of its NES-containing binding partners including APC, axin and others. Hence use of leptomycin B here will obfuscate any clear interpretation of the impact of IFT140 or klp64d knockdowns on beta-catenin. In addition, the nuclei shown in Fig. 5b and 5f look especially shrunken and bright - typical of chromatin condensation during altered cell cycle or apoptosis. Normally the nuclear envelope breaks down under these situations so staining of the nuclear rim is warranted.
5. Figure 6. The images in Figure 6f' suggest that IFT140 knockdown causes some reduced nuclear staining of ArmS10, and the images do not match the graphs shown in Supp data. In fact the practice of showing nuc/cyt ratios only for the fluorescence intensity measures is misleading as in many cases there is altered cytoplasmic levels of beta-catenin. It is suggested that the nuclear and cytoplasmic levels also be shown to clarify genuine changes. Fig 6g, show MBP alone as control for binding.
6. Figure 7. Some of the clearest data in the paper - it shows that loss of IFT140 in MEFs cause an increase in cytoplasmic levels of beta-catenin before and after wnt signaling. Most consistent with IFT140 either increasing nuclear retention or competing for some cytoplasmic retention of beta-catenin. This could readily be assessed by mild detergent extraction assays, as beta-catenin is weakly retained in the cytoplasm at many vesicular and organellar structures, including Golgi as previously suggested by the first author in reference 19.

General comments about the study design.

At most the authors can conclude that klp64d and IFT140 contribute to wnt signaling in drosophila, and can contribute in an undefined way to nuclear accumulation of beta-catenin in drosophila and mouse cells. L. Vuong already showed previously (Ref 19) that klp64d binds directly to beta-catenin arm repeats, co-locates with beta-catenin in cytoplasmic vesicles and also showed that human KIF3A binds to beta-catenin. Not clear how these findings fit with the partially-supported model outlined in Figure 3F. Moreover, KAP3 can bind both APC (Jimbo et al, NCB 2002) and Nup358 (Murawala et al JCS 2009) which themselves bind beta-catenin, and so there are potentially multiple ways in which beta-catenin could associate with these kinesin-2 subunits or with IFT140. Given that kinesin-2 complexes are involved in anterograde transport to the plasma membrane, it was not explained here, or substantiated in any way, how such interaction will stimulate MT-minus end directed transport to the nucleus. Even in the absence of a true motor function, the authors would need a substantive amount of extra data, and a proper read of the relevant literature, in order to show how IFT140 causes enhanced nuclear staining of beta-catenin.

Reviewer #2:

Remarks to the Author:

Major Strength:

The work is a comprehensive study, utilizing an impressive array of in vivo and in vitro approaches, suggesting a non-ciliary role for kinesin-2 and IFT-A proteins in the Wnt/Wg pathway in both drosophila and mice. The experiments are well designed, and include appropriate controls and statistical analyses. Evidence linking kinesin-2/IFT-A proteins to Wg signaling and target gene expression is convincing, with data clearly showing that kinesin-2 and IFT-A associate with Arm/ β -catenin, and are associated with its nuclear localization.

Major Weakness:

While the data clearly define a novel role for the kinesin2/IFTA complex in promoting Arm/ β -catenin transport, no insight is provided as to what that exact role is. The authors demonstrate the importance of kinesin-2 motor activity, but it is difficult to reconcile the activity of a plus-end directed motor with retrograde transport to the nucleus. An analogy is made to the role of another kinesin family member, KIF7, in the hedgehog signaling pathway, but that is also hard to reconcile as KIF7 acts to modify the microtubule architecture and is not involved in actual transport functions. Kinesin-2 is only known as a transport motor. Therefore, unless a novel function for kinesin-2 is being proposed and tested, it is difficult to make sense of the results – this is a major limitation of the paper.

Minor concerns:

-Page 12, line 256: Please explain mechanism of Leptomycin B as a nuclear export inhibitor upon first introduction.

-Page 14, line 279: A more rigorous explanation of how Klp64D mutants effect motor function is needed. For example, klp64D Δ ABD/klp64DT114A mutants are said to "fail to move along [microtubules]". Can they still bind to the microtubule? What are the mechanisms of these mutations that inhibits motor processivity?

-page16, line 342: "outlier" is misspelled.

-Page 19, line 393: While experiments do show that the requirement of IFT-A for nuclear translocation of β -catenin is conserved between Drosophila and mice, however it does not support

that this is conserved for all mammals.

-Page 28, line 614: Please confirm that step size is 0.5X the optical section thickness to satisfy the Nyquist theorem sampling criterion. For reference, Bolte and Cordelières (2006) "A guided tour into sub cellular colocalization analysis in light microscopy" Journal of Microscopy 224(3): 213-232 under "digital Imaging" subheading.

Reviewer #3:

Remarks to the Author:

In this manuscript Vuong et al expand on their previous work (Balmer et al 2015) and show, using *Drosophila* imaginal wing disc epithelium and salivary gland cells that kinesin 2 and IFT-A components facilitate Wg/Wnt signaling. They provide evidence that Arm/beta-catenin and Axin interact with a kinesin 2/klp68D-IFT-A complex. Moreover, they show that Wg/Wnt-mediated nuclear localization of Arm/beta-catenin is enabled by a functional kinesin 2-IFT-A complex. Interestingly, oncogenic ArmS10 mutant is insensitive to this regulation. They also show that a similar mechanism is present in MEF cells and that beta-catenin nuclear localization of beta-catenin is defective when IFT-A complex component IFT140, but not IFT-B complex component IFT88, is deleted. Overall most of the data are of high quality. However, a number of key experiments are missing to clarify certain ambiguities and to shed light on the mechanism under play as outlined below. These experiments are necessary to make the current manuscript a novelty in the field and suitable for publication in Nature Comm.

MAJOR COMMENTS:

1) While the data clearly shows that transcriptional targets of Wg, Sens and Dll are reduced because of Klp64D and Kap3 KD (Figure 1 incl. corresponding suppl. data), can they also show that Arm levels are unaffected in this tissue?

2) Where is there figure showing the data on klp68DRNAi? (line 145)

3) In Balmer et al (2015) klp68D (Kinesin 2 subunit Kif3B) knockdown seems not to have an effect on Sens expression. In the current manuscript, the authors state that klp68D depletion resulted in smaller wings and suggest that klp68D may have klp64D/Kap3-independent functions without mentioning its effect on Sens expression (data are not provided). The authors must give a better explanation to this discrepancy.

4) Data in Figure 2 provides strong evidence that a normal Klp64D expression is required for Wg- or AxnE77-induced Sens expression. However, there seems to be an undeniable reduction in the levels of Arm in the Ift122 and klp64D mutant clones. The authors do not explain this and it goes against their claim that "these results...indicate that they [kinesin 2/IFT-A complex] are required downstream of Arm stabilization in Wg/Wnt signaling" (lines 185-187). These data indicate that Klp64D and/or Ift22 have a general effect on Arm levels (compare Figure 2 a' with b' and c' as well as d' with e'). Clearly, Klp64D/Ift22 has an effect on Arm levels independently of the Wg signaling state. The authors need to provide a quantification of these data and give an explanation.

5) The authors have put in a tremendous amount of work in performing Co-IPs and pull-down assays to show that Arm (and Axin) interacts with IFT-A components IFT122 and IFT140 and that these components make direct contact with Kap3 subunit of kinesin 2 complex (Figure 3 + suppl. data). However, this is not sufficient proof that all these components are present in the same complex, and it is difficult to know the level of each protein component residing in such a complex. A size exclusion chromatography experiment would be important to address these queries. In addition, the authors will greatly improve their manuscript by showing these complexes are

dynamic and affected by Wg/Wnt signaling. Co-IP with IFT44 can serve as a negative control. Co-IP with Kif3B/klp68D would be a plus.

6)Figure S3 d-f: The authors should provide a WB of Arm from the same experiments.

7)Figure 3a: input blot shows absence of Arm in control cells. May be this reviewer is missing something here, but shouldn't Arm be expressed?

8)Can the authors explain what they mean by "exogenous Wg expression" (Figure 4 b – line 225), and clarify how this compares to the endogenous Wg expression? Can they show Wg expression in the cells by IFM to be sure that there is Wg expression?

9)Figure 4: there seems to be major co-localization between Klp64D-HA, IFT140-myc and Arm at AJs at the cell periphery. Could this mean that Klp64D/IFT140/Arm complexes are majorly present at AJs? Did the authors co-stain for cytoskeletal proteins in order to evaluate whether complexes are increased on e.g. MT tracks in + Wg cells?

10)This reviewer cannot find the "detailed assay description" (line 223).

11)Figure 4S: what are the dots? Are these cytoplasmic puncta or at the AJ as in Fig 4a? Where are the cell borders? As this reviewer understands it, the cells of the imaginal wing disc are very small and difficult to use for co-localization studies. This figure seems to be more confusing than informative.

12)Figure 5: overall very nice data. Complements for some of the issues raised above.

13)The authors state that "also of note, the increase in Arm/beta-catenin levels was variable from cell to cell, likely due to varying expression levels of the Gal4-driver" (line 269, Figure 5 + sup). Shouldn't the authors then use an antibody against Wg to show that e.g. nuclear localization of Arm occurs in cells with (high) expression of Wg? Can they also use extracts from these cells to show the efficiency of KD and o/e, since the antibodies do not work in IFM (line 219)?

14)Once again I see that Arm levels seem to be diminished in klp64D/ift140 KD (Fig 5 b, c as in Fig 2) and that with the over-expression of the klp64D motor mutant (Fig 5d,e) this effect is effectively reversed and Arm is highly stabilized (albeit cytoplasmic). It really seems that the klp64D and Ift140D promote Arm stability (independently of Axin activity) and that klp64D motor mutants act in a dominant positive fashion. Can the authors explain these results and provide WBs of Arm in lysates of these samples.

15)Figure S5: the authors should show a similar WB as in Figure S5-e for wild-type Arm and Arm*.

16)Figure 6 (+ suppl.): Overall very nice data. Complements for some of the issues addressed above.

17)Figure 7h: There is a significant reduction in the level of cytoplasmic beta-catenin in lane 1 compared to lane 3 (Wnt3a-CM-treated versus untreated cells). This is in contrast to published data that show cytoplasmic accumulation of beta-catenin upon Wnt/b-catenin activation (e.g. PMID: 25392450). How do the authors explain this? It is very likely that cytoplasmic beta-catenin is lost during the first step of fractionation (line 625). What they collect as cytoplasmic sample seems to this reviewer to be the plasma membrane-enriched fraction. They should also show total beta-catenin pool.

18)Figure 7h: What do the authors mean by "unconditioned media"? Is it just normal DMEM used in growing the MEFs (line 605)? They need to clarify this because normal media cannot serve as a

proper control.

19) Mammalian cells: The authors clearly show, by IFM, that beta-catenin is located to the nucleus after 24hr Wnt3a stimulation (Wnt3a-conditioned medium) and that this localization is lost in Ift140^{-/-} MEFs. This shows that IFT40 (and possibly IFT-A as a whole) plays a role in nuclear localization/maintenance of beta-catenin after Wnt activation. However, they do not show what happens at the transcriptional level. Is Wnt3a-induced Axin2 expression eliminated in Ift140^{-/-} MEFs? Are there Wnt/beta-catenin defects in Ift140^{-/-} embryos? It has also been shown that Kif3A is not required for normal Wnt/beta-catenin signaling in the developing mice (PMID: 19718259).

20) Was LepB used in the MEF experiments? Why/why not?

21) How do the primary cilia look in the MEF cells? How do the deletions affect the primary cilia structure?

22) What happens to ArmS10 and Arm* in mammalian cells? How is Kinesin 2-IFT-A-beta-catenin complex affected in response to Wnt stimulation?

MINOR COMMENTS

1) Figure 1 (+ suppl.): Can the authors provide WB evidence for the knockdowns as well as for overexpression of the IFT proteins? For example the reason IFT44 could not rescue the klp64D RNAi phenotype like the other tested IFT proteins could be due to lower expression levels.

2) Figures 1, S1, 2, S2, 5, S5 and 6: show scale bars.

3) Figure 3: the labelling of blots are very confusing. For example, what do the authors mean by labelling lane 2 in Fig S3d-f as GFP? Both lanes should contain GFP (Axin-GFP).

4) line 627: Surely the authors did not use 1M EDTA in their buffer. Please correct.

5) A schematic representation of their model (lines 457-483) would help of great help for the reader.

Reviewer #4:

Remarks to the Author:

In the present manuscript the authors describe a novel mechanism that allows β -catenin to translocate to the nucleus requiring Kinesin2/IFT-A complex, independently of the cilium. This mechanism appears to be conserved both in Drosophila and in mammals. They further postulate the existence of a yet unknown factor keeping Armadillo in the cytoplasm in the absence of Kinesin2/IFT-A.

The data presented here are strong and novel enough to be considered for publication in Nature Communications, but some issues might be addressed before accepting the manuscript.

Regarding the data/figures:

1. In Sentences 143-146 the authors state that the C96>klp68DRNAi causes a phenotype but they never show the data. Is there a reason for that? Either add the data or remove the statement.
2. It is not entirely clear how the authors explain the finding, that overexpression of IFT components can rescue the phenotype caused by loss of Klp64D. Aren't all those proteins supposed to be working in the same complex, as proposed in Fig 3f, albeit with different functions? How can the loss of one be compensated with a different one?
3. In all Figures with micrographs, I would suggest to the authors to include scale bars. In

addition, the authors should move the labeling letters outside of the wing area. Furthermore in figures g'-h'-i'-j' do the authors detect ectopic sens or is it just background? If the latter applies, then these panels should be replaced with better quality ones. Finally in panel i'' the authors claim that they observe restored Dll expression, that is not the case comparing a'' to i'', I would suggest to either replace the panel or change the statement in the text.

4. In SF1 move the genotypes over the wings like in Fig1 and move the labeling letters outside of the wing area.

5. Both on Figure 2 and SF2 remove the merge images, they don't add anything significant to the story and they are of very low quality.

6. In Figure 4 the point that the authors want to make is nicely supported by the data, the problem is that due to the need of high magnification the quality of the data is reduced. I would propose either to move Figure 4 to supplementary or maybe try to take new pictures with a lower magnification.

7. The data presented in figure 5 are not 100% in line with what is described in the text. In panel e'''-g'''-h'□ 019;' there seems to be some nuclear localization of Arm. The authors should add a quantification. It would also be interesting to see the effect of these different localizations of Arm on target genes. Would they also be activated in the situation where Arm is ubiquitous in the cell, or only when it is mostly nuclear?

8. Also in Figure 6 genotypes should be moved on top of the wings as in Figure1 and label letters should be out of the wing area.

9. In Figure 7 I think the monochrome immunostaining images should be either moved to supplementary or removed completely from the paper.

minor comments about the text:

1. The abstract is very hard to follow, does not give a clear and direct insight to the findings presented in the core text. In addition the authors repeat themselves in sentences 29-31.

2. At the end of line 46 a citation is needed.

3. It would be helpful to introduce what IFT-A and Klp64D is at the beginning of the section (line 61).

4. In line 66 Ref is a typo.

5. In line 139, discs not dics

6. In line 143 remove the word "very" because the data have a degree of similarity but not so striking as the authors imply.

7. In line 211 the correct form is Knock.

8. In the discussion part entitled "Kinesin 2/IFT-A complex promotes nuclear β -catenin localization" I think that from line 432 to 448 the text is again an introduction to Wnt signaling which exists already in the introduction of the manuscript. This part is offering no additional information to the reader and is also not fully relevant to the data discussed in that section.

POINT-BY-POINT RESPONSE to Reviewers' comments:

Reviewer #1 (Remarks to the Author):

The paper by Vuong et al proposes that a kinesin-2/IFT complex can stimulate wnt signaling through direct binding to beta-catenin and by promoting its translocation to the nucleus through a microtubule motor dependent function - and that this occurs in drosophila and mouse embryo fibroblasts. I found certain sections of this paper to be excellent, with a very thorough genetic dissection of the interactions between the kinesin-2 and IFT-A complex components. They present reasonable evidence that kinesin-2 (ie. klp64d) and the ciliary factor IFT-140 can influence nuclear levels of beta-catenin under specific circumstances following wnt stimulus. However there is no evidence in this paper to show that klp64d or IFT140 affect the rate of nuclear transport. The authors performed no actual cell assays to analyse nuclear import or export, or even standard CSK extraction assays to test for changes in nuclear or cytoplasmic retention of beta-catenin. The main conclusion to be drawn from the limited data shown is that there is some effect on nuclear beta-catenin, most likely due to altered retention in the cytoplasm. This is interesting, but without a mechanism it truly lacks impact.

We thank the reviewer for their thoughtful comments and assessing our paper as interesting. We have addressed all their comments through additional experiments as outlined below.

Major comments about Figures:

1. Figures 1 and 2 look okay.

Thank you for the positive comment.

Also, we have adjusted the figure labeling and lay-out to match the new Nature Communications presentation style

2. Figure 3f model. This diagram is partly supported by the data but there are some issues. (a) include an MBP-only control in Figure 3B, as beta-catenin is a promiscuous protein and it is important to exclude non-specific interactions. Other MBP-fusions do not directly address this issue.

We thank the reviewer for their suggestion and we have repeated and redone many of the experiments shown in Figure 3 with more complete controls. We used and included MBP-IFT144 protein as negative control, as it is related to IFT140, yet does not participate genetically and functionally in the process we're studying.

(b) The sentence in lines 205-207 suggest there is data showing that KAP3 and klp64d do not bind to beta-catenin in the Supp Fig 3, however I could see no such data presented.

Klp64D directly binds to beta-catenin, albeit weaker (this was shown in a previous paper; Vuong et al, Development, 2014). Kap3 directly binds only to IFT140 (as shown in the revised Fig 3e), it does not bind to beta-catenin (as was also shown in Vuong et al, Development, 2014). Again these figures were redone and a summary model, based on the data shown in this paper and from previous work, is presented in panel 3f.

(c) In addition, this statement contradicts the finding previously published by the first author, L. Vuong in Ref 19, showing that klp64D binds directly to beta-catenin via its Armadillo repeat domain.

As mentioned above, Klp64D (but not Kap3) directly binds to beta-catenin. We trust that the revised figure and text now explain this adequately.

3. Figure 4. Co-staining of ectopic klp64d-HA and IFT140-Myc. The cell images look overexposed, obviously to boost signal. But the co-staining looks sporadic and such images need some

quantification to demonstrate the extent of co-localization in a larger population of cells (at least 50-100).

We thank the reviewer for their helpful suggestion. We have redone this figure, showing a larger field of cells and have added quantifications (panel 4i) and also added an xz-plane to show co-localization in the subapical region (new panels 4e-h). We have analyzed many salivary glands of all genotypes and the results are very reproducible. It is now also well documented in the staining panels and the quantifications that the co-localization is fully dependent on Wg-signaling activation (+Wg genotypes). The text has been modified accordingly.

Also in Supp Fig 4d, there are blobs of IFT140/Arm co-staining that look less like specific "puncta" and more like artificial aggregates caused by protein overexpression.

We thank the reviewer for their constructive comment. We have modified Fig S4 to better document the in vivo co-staining and also indicated where in the disc (near the Wg expressing stripe at the D/V-boundary) we are imaging. We trust that the new revised Figure S4 better documents the localization studies and co-localizations.

4. Figure 5. The authors used larger salivary glands to detect nuclear staining, however they employed use of the CRM1 inhibitor leptomycin B (normally abbreviated LMB) to boost the visible nuclear beta-catenin levels after wnt stimulation. As shown in the previous (but mostly uncited) literature, beta-catenin is a shuttling protein and its nuclear localization can be increased indirectly by LMB through forced retention of its NES-containing binding partners including APC, axin and others. Hence use of leptomycin B here will obfuscate any clear interpretation of the impact of IFT140 or klp64d knockdowns on beta-catenin. In addition, the nuclei shown in Fig. 5b and 5f look especially shrunken and bright - typical of chromatin condensation during altered cell cycle or apoptosis. Normally the nuclear envelope breaks down under these situations so staining of the nuclear rim is warranted.

We appreciate the concern of the reviewer and we show now different samples of the respective genotypes, where nuclear appearance always resembles wild-type and is the same in all genotypes analyzed. Importantly, we would like to emphasize that all these experiments are very well controlled and we only see nuclear Arm localization in the presence of Wg, and hence activation of Wg-signaling is required for Arm nuclear translocation. We do not see any nuclear Arm in leptomycin B treated salivary glands that were not exposed to Wg, which serves as a proper control.

5. Figure 6. The images in Figure 6f suggest that IFT140 knockdown causes some reduced nuclear staining of ArmS10, and the images do not match the graphs shown in Supp data. In fact the practice of showing nuc/cyt ratios only for the fluorescence intensity measures is misleading as in many cases there is altered cytoplasmic levels of beta-catenin. It is suggested that the nuclear and cytoplasmic levels also be shown to clarify genuine changes. Fig 6g, show MBP alone as control for binding.

We would like to emphasize that in the original Fig6f it was Sens staining (red channel in new Fig 6f), and it was/is not nuclear staining of ArmS10. We have added controls to the new panel 6l (formerly 6g), showing MBP-IFT144 as a negative control. We have also added new and improved images for the respective salivary glands in the panels 6j and 6k, to better document the differences between the two stable ArmS10 and Arm* isoforms.

6. Figure 7. Some of the clearest data in the paper - it shows that loss of IFT140 in MEFs cause an increase in cytoplasmic levels of beta-catenin before and after wnt signaling. Most consistent with IFT140 either increasing nuclear retention or competing for some cytoplasmic retention of beta-catenin. This could readily be assessed by mild detergent extraction assays, as beta-catenin is weakly

retained in the cytoplasm at many vesicular and organellar structures, including Golgi as previously suggested by the first author in reference 19.

We thank the reviewer for their praise of this figure. We trust that this is a very well controlled assay, as we use a IFT-B component knock-down (*ift88^{-/-}*) as specificity control for an effect of IFT-A.

In the revised paper, we have now also added new data to the revised Figure 7 that document that effects seen with the mouse beta-catenin activated isoforms, analogous to ArmS10 and Arm*, behave exactly the same as ArmS10 and Arm* in *Drosophila*, showing the very same different behavior vis-à-vis an IFT-A or kinesin-2 requirement. These new data further strengthen the conclusions and confirm that the existing mechanism is fully conserved between *Drosophila* and mouse/mammals.

General comments about the study design.

At most the authors can conclude that *klp64d* and IFT140 contribute to wnt signaling in *drosophila*, and can contribute in an undefined way to nuclear accumulation of beta-catenin in *drosophila* and mouse cells. L. Vuong already showed previously (Ref 19) that *klp64d* binds directly to beta-catenin arm repeats, co-locates with beta-catenin in cytoplasmic vesicles and also showed that human KIF3A binds to beta-catenin. Not clear how these findings fit with the partially-supported model outlined in Figure 3F. Moreover, KAP3 can bind both APC (Jimbo et al, NCB 2002) and Nup358 (Murawala et al JCS 2009) which themselves bind beta-catenin, and so there are potentially multiple ways in which beta-catenin could associate with these kinesin-2 subunits or with IFT140. Given that kinesin-2 complexes are involved in anterograde transport to the plasma membrane, it was not explained here, or substantiated in any way, how such interaction will stimulate MT-minus end directed transport to the nucleus. Even in the absence of a true motor function, the authors would need a substantive amount of extra data, and a proper read of the relevant literature, in order to show how IFT140 causes enhanced nuclear staining of beta-catenin.

The reviewer raises an important point about directionality of the protein complex Kinesin-2/IFTA along cytoplasmic microtubules. However, this concern is essentially based on the assumption that cytoplasmic microtubules interact with the nuclear membrane solely through **their minus ends**. Although it was reported that the minus end of microtubules can interact with the nuclear envelop via non-centrosomal subsets of gamma-tubulin molecules, recent studies on protein complexes linking the nucleus to components of the cytoskeleton, including microtubules, suggest alternative scenarios. The linker of nucleoskeleton and cytoskeleton (LINC) complex, composed of outer and inner nuclear membrane Klarsicht, ANC-1, and Syne homology (KASH) and Sad1 and UNC-84 (SUN) proteins, respectively, connects the nucleus to cytoskeletal filaments and performs diverse functions including nuclear positioning, mechanotransduction, and meiotic chromosome movements. The KASH proteins expressed allows for binding to actin and microtubules (Gundersen and Worman, 2013 and Chang et al. 2015). In most cases, the interaction with microtubules is mediated through association of the KASH protein with the motor proteins, kinesin, dynein, or both. This interaction occurs along the length of microtubules not excluding their plus ends (Zhu et al, 2017).

We agree with the reviewer that the presumed organization of MTs is confusing given our findings, but I would like to emphasize that in non-dividing epithelial cells most MTs are not organized by the centrosomes and thus it cannot be assumed we know how MTs are oriented. In PCP studies in *Drosophila*, for example, it was demonstrated that non-centriole based MTs are the majority in the respective epithelial cells and that they are stochastically oriented, with maybe a bias in the proximal-distal axis in the subapical region (several papers from the Axelrod and Uemura labs). There are also several studies in cell culture based assays that suggest the organization and orientation of microtubules is more complex in most cell types (see for example Zhu, Liu, and Gundersen, *Seminars in Cell & Dev Biol*, (2017) "Nuclear positioning in migrating fibroblasts"; or Rowning et al, *PNAS*, 1997, Vol 94, 1224-1229). I trust that the reviewer will agree with us that a detailed understanding of MT

organization and orientation in the respective epithelial cells and MEFs is a full new project and clearly outside the scope of this study. We have agreed with the editor that we cannot address a full mechanism in this paper.

We have addressed all specific suggestions, as outlined above, with many additional experiments that significantly strengthen the conclusions. We trust that the paper is significantly improved with all the new data and additions.

--

Reviewer #2 (Remarks to the Author):

Major Strength:

The work is a comprehensive study, utilizing an impressive array of in vivo and in vitro approaches, suggesting a non-ciliary role for kinesin-2 and IFT-A proteins in the Wnt/Wg pathway in both drosophila and mice. The experiments are well designed, and include appropriate controls and statistical analyses. Evidence linking kinesin-2/IFT-A proteins to Wg signaling and target gene expression is convincing, with data clearly showing that kinesin-2 and IFT-A associate with Arm/ β -catenin, and are associated with its nuclear localization.

We are grateful to the reviewer for pointing out the strengths and praising our data as “clear” and “convincing”.

Major Weakness:

While the data clearly define a novel role for the kinesin2/IFTA complex in promoting Arm/ β -catenin transport, no insight is provided as to what that exact role is. The authors demonstrate the importance of kinesin-2 motor activity, but it is difficult to reconcile the activity of a plus-end directed motor with retrograde transport to the nucleus. An analogy is made to the role of another kinesin family member, KIF7, in the hedgehog signaling pathway, but that is also hard to reconcile as KIF7 acts to modify the microtubule architecture and is not involved in actual transport functions. Kinesin-2 is only known as a transport motor. Therefore, unless a novel function for kinesin-2 is being proposed and tested, it is difficult to make sense of the results – this is a major limitation of the paper.

We thank the reviewer for highlighting the novelty of our discovery of IFT-A/Kinesin-2 function in Wnt/beta-catenin signaling.

We agree with the reviewer that the presumed organization of MTs is confusing given our findings, but I would like to emphasize that in non-dividing epithelial cells most MTs are not organized by the centrioles and thus it cannot be assumed we know how MTs are oriented. In PCP studies in *Drosophila*, for example, it was demonstrated that non-centriole based MTs are the majority in the respective epithelial cells and that they are stochastically oriented, with maybe a bias in the proximal-distal axis in the subapical region (several papers from the Axelrod and Uemura labs). There are also several studies in cell culture based assays that suggest the organization and orientation of microtubules is more complex in most cell types (see for example Zhu, Liu, and Gundersen, *Seminars in Cell & Dev Biol*, (2017) “Nuclear positioning in migrating fibroblasts”; or Rowning et al, *PNAS*, 1997, Vol 94, 1224-1229). I trust that the reviewer will agree with us that a detailed understanding of MT organization and orientation in the respective epithelial cells and MEFs is a full project and clearly outside the scope of this study

Minor concerns:

-Page 12, line 256: Please explain mechanism of Leptomycin B as a nuclear export inhibitor upon first introduction.

We thank the reviewer for pointing out this omission. We have added this in the revised text accordingly on pg 13 and also in the respective Figure 5 legend.

-Page 14, line 279: A more rigorous explanation of how Klp64D mutants effect motor function is needed. For example, klp64D Δ ABD/klp64DT114A mutants are said to “fail to move along [microtubules]”. Can they still bind to the microtubule? What are the mechanisms of these mutations that inhibits motor processivity?

We have added explanatory text to better explain the nature of these mutant isoforms on page 14.

-page16, line 342: “outlier” is misspelled.

Thanks, this has been corrected.

-Page 19, line 393: While experiments do show that the requirement of IFT-A for nuclear translocation of β -catenin is conserved between Drosophila and mice, however it does not support that this is conserved for all mammals.

We appreciate the concern of the reviewer, and we state now specifically that it is conserved between Drosophila and mice, and thus likely also in other mammals.

-Page 28, line 614: Please confirm that step size is 0.5X the optical section thickness to satisfy the Nyquist theorem sampling criterion. For reference, Bolte and Cordelières (2006) “A guided tour into sub cellular colocalization analysis in light microscopy” Journal of Microscopy 224(3): 213-232 under “digital Imaging” subheading.

We apologize for not having stated this; we confirm the proper application as it is standard within our Microscopy Core facility and supported by their staff. This is now mentioned in the Methods.

--

Reviewer #3 (Remarks to the Author):

In this manuscript Vuong et al expand on their previous work (Balmer et al 2015) and show, using Drosophila imaginal wing disc epithelium and salivary gland cells that kinesin 2 and IFT-A components facilitate Wg/Wnt signaling. They provide evidence that Arm/beta-catenin and Axin interact with a kinesin 2/klp68D-IFT-A complex. Moreover, they show that Wg/Wnt-mediated nuclear localization of Arm/beta-catenin is enabled by a functional kinesin 2-IFT-A complex. Interestingly, oncogenic ArmS10 mutant is insensitive to this regulation. They also show that a similar mechanism is present in MEF cells and that beta-catenin nuclear localization of beta-catenin is defective when IFT-A complex component IFT140, but not IFT-B complex component IFT88, is deleted. Overall most of the data are of high quality. However, a number of key experiments are missing to clarify certain ambiguities and to shed light on the mechanism under play as outlined below. These experiments are necessary to make the

current manuscript a novelty in the field and suitable for publication in Nature Comm.

We thank the reviewer for praising the overall strengths of our paper and the clarity of the data. We are grateful for their constructive suggestions on how to improve the paper further.

We are pleased to say that we were able to fully complete all suggested additional experiments suggested, as outlined below.

MAJOR COMMENTS:

1)While the data clearly shows that transcriptional targets of Wg, Sens and Dll are reduced because of Klp64D and Kap3 KD (Figure 1 incl. corresponding suppl. data), can they also show that Arm levels are unaffected in this tissue?

We have added a Western blot analyzing Arm expression/levels to the revised Suppl. FigS5 (new panel S5e).

2)Where is there figure showing the data on klp68DRNAi? (line 145)

We apologize for this omission. We have added the respective panels on Klp68D to the revised Suppl. FigS1.

3)In Balmer et al (2015) klp68D (Kinesin 2 subunit Kif3B) knockdown seems not to have an effect on Sens expression. In the current manuscript, the authors state that klp68D depletion resulted in smaller wings and suggest that klp68D may have klp64D/Kap3-independent functions without mentioning its effect on Sens expression (data are not provided). The authors must give a better explanation to this discrepancy.

We have added a new panel on the effects or lack thereof on Sens expression of Klp68D (including staining with anti-Sens in the wing pouch) to Suppl. FigS1.

4)Data in Figure 2 provides strong evidence that a normal Klp64D expression is required for Wg- or AxnE77-induced Sens expression. However, there seems to be an undeniable reduction in the levels of Arm in the lft122 and klp64D mutant clones. The authors do not explain this and it goes against their claim that “these results...indicate that they [kinesin 2/IFT-A complex] are required downstream of Arm stabilization in Wg/Wnt signaling” (lines 185-187). These data indicate that Klp64D and/or lft22 have a general effect on Arm levels (compare Figure 2 a’ with b’ and c’ as well as d’ with e”). Clearly, Klp64D/lft22 has an effect on Arm levels independently of the Wg signaling state. The authors need to provide a quantification of these data and give an explanation.

We thank the reviewer for their helpful comment. To address this we have added a new Western Blot analyzing Arm expression/levels in the respective genetic conditions to the revised Suppl Fig S5, new panel S5e).

5)The authors have put in a tremendous amount of work in performing Co-IPs and pull-down assays to show that Arm (and Axin) interacts with IFT-A components IFT122 and IFT140 and that these components make direct contact with Kap3 subunit of kinesin 2 complex (Figure 3 + suppl. data). However, this is not sufficient proof that all these components are present in the same complex, and it is difficult to know the level of each protein component residing in such a complex. A size exclusion chromatography experiment would be important to address these queries. In addition, the authors will greatly improve their manuscript by showing these complexes are dynamic and affected by Wg/Wnt signaling. Co-IP with IFT44 can serve as a negative control. Co-IP with Kif3B/klp68D would be a plus.

We thank the reviewer for their suggestion to use IFT144 as a negative control. We have added IFT144 as a control to all CoIPs and pull down experiments now shown in the revised Figure 3.

6)Figure S3 d-f: The authors should provide a WB of Arm from the same experiments.

We have added a Co-IP with Arm to the revised Suppl. FigS3, panels g-h.

7)Figure 3a: input blot shows absence of Arm in control cells. May be this reviewer is missing something here, but shouldn't Arm be expressed?

We thank the reviewer for pointing this out. In the revised Figure 3, a new panel Fig 3a is now shown which contains all respective controls.

8)Can the authors explain what they mean by “exogenous Wg expression” (Figure 4 b – line 225), and clarify how this compares to the endogenous Wg expression? Can they show Wg expression in the cells by IFM to be sure that there is Wg expression?

We refer to as “exogenous” to expression that is driven and induced by the Gal4/UAS-system and we use here with a salivary gland specific driver. This approach allows the comparison of Wg-induced or Wg-independent co-localization; for example IFT-A and Arm only colocalize upon Wg-signaling, whereas Kinesin 2 and IFT-A colocalize all the time. We can thus deduce how activation of Wg-signaling affects Arm/beta-catenin localization/colocalization. We trust we have now explained this better in the text.

9)Figure 4: there seems to be major co-localization between Klp64D-HA, IFT140-myc and Arm at AJs at the cell periphery. Could this mean that Klp64D/IFT140/Arm complexes are majorly present at AJs? Did the authors co-stain for cytoskeletal proteins in order to evaluate whether complexes are increased on e.g. MT tracks in + Wg cells?

This is a very good point, thank you. We have now added tubulin staining to the “all new” Figure 4, which addresses this point. Co-localization of Tubulin and Arm is observed, but only in cells where Wg-signaling is activated (+Wg salivary gland cells). This is added as panels 4c,d and quantified in 4i in the new Fig 4.

10)This reviewer cannot find the “detailed assay description” (line 223).

We apologize for this confusion. The assay description is in the subsequent section under subheading “The Kinesin-2/IFT-A complex is required for nuclear localization of Arm” (specifically on lines 253-277). We have corrected the statement in the text.

11)Figure 4S: what are the dots? Are these cytoplasmic puncta or at the AJ as in Fig 4a? Where are the cell borders? As this reviewer understands it, the cells of the imaginal wing disc are very small and difficult to use for co-localization studies. This figure seems to be more confusing than informative.

We have redone figure 4, showing a larger field of cells and have added quantification (panel 4i) and also added an xz-plane to show co-localization (new panels 4e-h). Accordingly, we have also redone Fig S4, indicating now which area of the disc is being shown (near the Wg-expression domain at the D/V boundary) and we have added an xz-plane for the respective stainings. We trust that the combination of the new Fig 4 and new Fig S4 clarifies and better documents the relevant issues.

12)Figure 5: overall very nice data. Complements for some of the issues raised above.

Thank you for this very nice and positive comment.

13)The authors state that “also of note, the increase in Arm/beta-catenin levels was variable from cell to cell, likely due to varying expression levels of the Gal4-driver” (line 269, Figure 5 + sup). Shouldn't the authors then use an antibody against Wg to show that e.g. nuclear localization of Arm occurs in cells with (high) expression of Wg? Can they also use extracts from these cells to show the efficiency of KD and o/e, since the antibodies do not work in IFM (line 219)?

We have added knock-down data of Axin and Klp64D (stained for Arm and DE-cad) to the revised Suppl. FigS5. Both in the revised Fig 5 and Fig S5, the levels of Arm stabilization are very apparent in Axin knock-down salivary glands, documenting some of the mosaicism and also showing cell-autonomy of the IFT-A/Kinesin 2 requirement. We trust that the expression of Wg is similar, but as Wg is secreted its effects are more regional.

14)Once again I see that Arm levels seem to be diminished in klp64D/ift140 KD (Fig 5 b, c as in Fig 2) and that with the over-expression of the klp64D motor mutant (Fig 5d,e) this effect is effectively reversed and Arm is highly stabilized (albeit cytoplasmic). It really seems that the klp64D and Ift140D promote Arm stability (independently of Axin activity) and that klp64D motor mutants act in a dominant

positive fashion. Can the authors explain these results and provide WBs of Arm in lysates of these samples.

Thank you for pointing this out. We have added a Western Blot analyzing Arm levels in all these genetic combinations to the revised Fig S5, and thus it is now well documented. On the Western Blot there appears to be indeed more Arm present in the klp64D motor mutants. This is now also discussed in the text.

15)Figure S5: the authors should show a similar WB as in Figure S5-e for wild-type Arm and Arm*. This has also been added to the new Western Blot in the revised Suppl Fig S5, panel S5e.

16)Figure 6 (+ suppl.): Overall very nice data. Complements for some of the issues addressed above. We are grateful to the reviewer for this very nice and positive comment.

17)Figure 7h: There is a significant reduction in the level of cytoplasmic beta-catenin in lane 1 compared to lane 3 (Wnt3a-CM-treated versus untreated cells). This is in contrast to published data that show cytoplasmic accumulation of beta-catenin upon Wnt/b-catenin activation (e.g. PMID: 25392450). How do the authors explain this? It is very likely that cytoplasmic beta-catenin is lost during the first step of fractionation (line 625). What they collect as cytoplasmic sample seems to this reviewer to be the plasma membrane-enriched fraction. They should also show total beta-catenin pool.

We appreciate the comment and concern of the reviewer. We have added explanatory text to revised Figure and Figure legend, see also below point 18 for additional controls.

18)Figure 7h: What do the authors mean by “unconditioned media”? Is it just normal DMEM used in growing the MEFs (line 605)? They need to clarify this because normal media cannot serve as a proper control.

We apologize for this confusion. We were using control conditioned media from L-cells that were treated the same way but transfected with the vector expressing GFP rather than Wnt3a (now this control is referred to as L-cells). We have corrected this in the text and refer to it as “control conditioned media”. We have also added an additional control media, one conditioned with Wnt5a, which does not induce the canonical Wnt-signaling response in these cells. The new blot with all these control media is shown in the revised Figure 7, panel j.

19)Mammalian cells: The authors clearly show, by IFM, that beta-catenin is located to the nucleus after 24hr Wnt3a stimulation (Wnt3a-conditioned medium) and that this localization is lost in Ift140^{-/-} MEFs. This shows that IFT40 (and possibly IFT-A as a whole) plays a role in nuclear localization/maintenance of beta-catenin after Wnt activation. However, they do not show what happens at the transcriptional level. Is Wnt3a-induced Axin2 expression eliminated in Ift140^{-/-} MEFs? Are there Wnt/beta-catenin defects in Ift140^{-/-} embryos? It has also been shown that Kif3A is not required for normal Wnt/beta-catenin signaling in the developing mice (PMID: 19718259).

We appreciate the reviewer’s insight and suggestion. We have added the transcriptional activation of Axin2 expression as a Wnt-signaling reporter and read-out to the MEF experiments. This is now shown in the new Suppl. Fig. S7.

20)Was LepB used in the MEF experiments? Why/why not?

LepB was not used in the MEF experiments as beta-catenin was readily detected in the nucleus in the Wnt3a induced conditions. This is in contrast to Drosophila, where Arm historically has actually not been seen in any nuclei in the past. Thus the LepB treatment was necessary to visualize its nuclear localization there. This is now clarified in the figure legends.

21)How do the primary cilia look in the MEF cells? How do the deletions affect the primary cilia structure?

Thank you for this very helpful suggestion. We have had these data in hand and we have now added them in the new Suppl Fig S7. In both, *ift140*^{-/-} and *ift88*^{-/-} mutant MEFs the cilia are basically lost or very short (see Fig S7 panels). As such we can conclude that the presence or absence of cilia has no effect on Wnt-signaling induced beta-catenin nuclear localization, as in *ift88*^{-/-} cells Wnt3a induces nuclear beta-catenin localization indistinguishably from wild-type.

22)What happens to ArmS10 and Arm* in mammalian cells?

We thank the reviewer for suggesting this additional data set. It took a lot of extra work, but we agree that it was worth it.

The behavior of the respective ArmS10 and Arm* mutations in mouse beta-catenin in MEFs completely mirrored the behavior of the Drosophila mutant isoforms in Drosophila. This new data set was added to the revised Fig 7, new panels 7k-7n. It is striking how similar the behavior is and thus it confirms that the general mechanism of IFT-A function in Wnt–signaling should be highly conserved across species.

MINOR COMMENTS

1)Figure 1 (+ suppl.): Can the authors provide WB evidence for the knockdowns as well as for overexpression of the IFT proteins? For example the reason IFT44 could not rescue the *klp64D* RNAi phenotype like the other tested IFT proteins could be due to lower expression levels.

We thank the reviewer for their helpful suggestion. We have added a Western blot to Suppl Fig S1 that documents the knock down levels of *Klp64d* in these genotypes, and the overexpression level of IFT144, as compared to IFT140 and IFT122, with all three IFT-A factors being expressed at very comparable levels.

2)Figures 1, S1, 2, S2, 5, S5 and 6: show scale bars.

Thank you for pointing this out. We have added scale bars to all figures.

3)Figure 3: the labelling of blots are very confusing. For example, what do the authors mean by labelling lane 2 in FigS3d-f as GFP? Both lanes should contain GFP (Axin-GFP).

We apologize for the potential confusion. We have revised and redone both, Figure 3 and Figure S3, also with additional controls where appropriate. We trust that the figures are now improved and clearer.

4)line 627: Surely the authors did not use 1M EDTA in their buffer. Please correct.

Thank you, this has been corrected; it should say 1mM.

5)A schematic representation of their model (lines 457-483) would help of great help for the reader.

We have edited the text to make it clearer, we trust that it is now easier to follow.

--

Reviewer #4 (Remarks to the Author):

In the present manuscript the authors describe a novel mechanism that allows β -catenin to translocate to the nucleus requiring Kinesin2/IFT-A complex, independently of the cilium. This

mechanism appears to be conserved both in *Drosophila* and in mammals. They further postulate the existence of a yet unknown factor keeping Armadillo in the cytoplasm in the absence of Kinesin2/IFT-A.

The data presented here are strong and novel enough to be considered for publication in *Nature Communications*, but some issues might be addressed before accepting the manuscript.

We are grateful to the reviewer for praising our data as “strong and “novel”, and highlighting the overall strengths and clarity of the data, and recommending publication in *Nature Communications*. We thank the reviewer for their constructive suggestions on how to improve the paper further and we have addressed all their comments as outlined below.

Regarding the data/figures:

1. In Sentences 143-146 the authors state that the C96>klp68DRNAi causes a phenotype but they never show the data. Is there a reason for that? Either add the data or remove the statement.

We have added the respective data on Klp68D to the revised Suppl. FigS1.

2. It is not entirely clear how the authors explain the finding, that overexpression of IFT components can rescue the phenotype caused by loss of Klp64D. Aren't all those proteins supposed to be working in the same complex, as proposed in Fig 3f, albeit with different functions? How can the loss of one be compensated with a different one?

We analyze the effect of over-expression of the respective components in a knock-down background of Klp64D (it is not a loss of the protein, only its reduction). As such, providing more a complex partner often can alleviate the effect of the knock-down/reduction, and this is what we see with the specific IFT-A complex components, but not with the equivalent controls.

3. In all Figures with micrographs, I would suggest to the authors to include scale bars. In addition, the authors should move the labeling letters outside of the wing area. Furthermore in figures g'-h'-i'-j' do the authors detect ectopic sens or is it just background? If the latter applies, then these panels should be replaced with better quality ones. Finally in panel i" the authors claim that they observe restored Dll expression, that is not the case comparing a" to i", I would suggest to either replace the panel or change the statement in the text.

We apologize for having omitted the scale bars. Scale bars have now been added to all Figures. We have also replaced panel 1i with a more representative wing disc picture.

4. In SF1 move the genotypes over the wings like in Fig1 and move the labeling letters outside of the wing area.

We thank the reviewer for pointing this out. We have adjusted the presentation of the genotypes and panel labels accordingly, to mirror also how it is presented in the main Figure 1.

5. Both on Figure 2 and SF2 remove the merge images, they don't add anything significant to the story and they are of very low quality.

We appreciate the comment of the reviewer. We have however decided to keep the merged channel panels for completeness and also after consulting with several colleagues. I trust that the reviewer will concur with us.

6. In Figure 4 the point that the authors want to make is nicely supported by the data, the problem is that due to the need of high magnification the quality of the data is reduced. I would propose either to move Figure 4 to supplementary or maybe try to take new pictures with a lower magnification.

We thank the reviewer for their helpful suggestion. We have redone Figure 4 completely and show now lower magnification of overall salivary glands, with high magnification insets. We have also added xz-plane micrographs and quantifications to best document the Wg-induced co-localizations.

7. The data presented in figure 5 are not 100% in line with what is described in the text. In panel e''-g''-h'' there seems to be some nuclear localization of Arm. The authors should add a quantification. It would also be interesting to see the effect of these different localizations of Arm on target genes. Would they also be activated in the situation where Arm is ubiquitous in the cell, or only when it is mostly nuclear?

We have rephrased this in the text to make the respective statement as accurate as possible. We trust that ubiquitous Arm localization would induce target genes (as a reminder: in *Drosophila* imaginal discs cells nuclear Arm is never detected as it is presumably below detection levels and yet target genes are being activated).

8. Also in Figure 6 genotypes should be moved on top of the wings as in Figure 1 and label letters should be out of the wing area.

Thanks for the suggestion, we have adjusted the figure labels.

9. In Figure 7 I think the monochrome immunostaining images should be either moved to supplementary or removed completely from the paper.

We appreciate the comment, but we have kept the monochrome images in the figure, as when we presented the data to other people, several suggested to add the monochrome panels for clarity.

minor comments about the text:

1. The abstract is very hard to follow, does not give a clear and direct insight to the findings presented in the core text. In addition the authors repeat themselves in sentences 29-31.

We thank their reviewer for their helpful comment. We have edited the Abstract and trust it is now easier to read and follow.

2. At the end of line 46 a citation is needed.

Thank you for the constructive comment, we have fixed this issue.

3. It would be helpful to introduce what IFT-A and Klp64D is at the beginning of the section (line 61).

We have added references and introductory remarks to this end in the respective paragraph.

4. In line 66 Ref is a typo.

Thank you for pointing this out. It has been corrected.

5. In line 139, discs not dics

This has been corrected.

6. In line 143 remove the word "very" because the data have a degree of similarity but not so striking as the authors imply.

This has been adjusted and "very" was removed.

7. In line 211 the correct form is Knock.

This was corrected.

8. In the discussion part entitled “Kinesin 2/IFT-A complex promotes nuclear β -catenin localization” I think that from line 432 to 448 the text is again an introduction to Wnt signaling which exists already in the introduction of the manuscript. This part is offering no additional information to the reader and is also not fully relevant to the data discussed in that section.

Thank you for pointing this out. We have removed the irrelevant “introduction to Wnt signaling” text from the Discussion.

Reviewers' Comments:

Reviewer #1:

Remarks to the Author:

The authors have performed substantial new experiments which satisfy my previous concerns and strengthen the conclusions of the paper.

Reviewer #3:

Remarks to the Author:

This revised version of the manuscript by Vuong et al is very much improved, and it is near acceptance for publication in Nature Communications.

However, there are a few issues left that need to be addressed before final acceptance can be made:

Re 4) The authors now point out that Arm levels decrease in the mutant clones for *klp64D* and *ift22* (Fig2). They should give a brief explanation about this (maybe refer to Fig S5) and make it very clear why they conclude that these factors act downstream of beta-catenin stabilization and not upstream (or both). This will greatly clarify the text.

Re 5) The addition of IFT144 is a major improvement to the results, and the Co-IP and pull-down data are of very high quality (Fig3 and Fig S3). However, I think it is reasonable to suggest that the authors should rearticulate their conclusion, such that data obtained using Co-IP and pull-down in conjunction with the genetic associations only favors the conclusion that the proteins exist in a IFT-A/kinesin-2 architecture (lines 217-219) after Wg stimulation.

Re 17) The authors insist on calling what looks to this referee to be membrane-enriched fraction as "cytoplasmic fraction". The cytoplasmic fraction is lost during the mincing in the PBS step (as explained in the method section), and the literature indicates cytoplasmic accumulation of beta-catenin upon Wnt3a stimulation (not seen in Fig 7j). I also do not find any clarification of this in the text or the legend. The authors can explain this or remove the panels for "cytoplasmic fraction" and replace them with total cell extract to show total beta-catenin levels for comparison with the nuclear fraction. For cytoplasmic fraction of beta-catenin the authors can use a hypotonic buffer or a saponin/digitonin-based buffer (see for example PMID: 29977599 and PMID: 25392450), but this is not required since the work is focused on nuclear localization of beta-catenin.

Other minor points:

Figure 4: Indicate in the figure if panels e-h are -Wg or +Wg.

Line 1005: legend for figure 7 (change "mote" to "note")

POINT-BY-POINT RESPONSE to Final Reviewers' comments:

Reviewer #1 (Remarks to the Author):

The authors have performed substantial new experiments which satisfy my previous concerns and strengthen the conclusions of the paper.

Thank you for the positive assessment of our revisions. No changes needed.

--

Reviewer #3 (Remarks to the Author):

This revised version of the manuscript by Vuong et al is very much improved, and it is near acceptance for publication in Nature Communications.

Thank you for the positive assessment of our revisions.

However, there are a few issues left that need to be addressed before final acceptance can be made:

Re 4) The authors now point out that Arm levels decrease in the mutant clones for klp64D and ift22 (Fig2). They should give a brief explanation about this (maybe refer to Fig S5) and make it very clear why they conclude that these factors act downstream of beta-catenin stabilization and not upstream (or both). This will greatly clarify the text.

We have added the respective statements about Arm levels in the paragraph describing the data (lines 154-162), and also in the respective section in the Discussion.

Re 5) The addition of IFT144 is a major improvement to the results, and the Co-IP and pull-down data are of very high quality (Fig3 and Fig S3). However, I think it is reasonable to suggest that the authors should rearticulate their conclusion, such that data obtained using Co-IP and pull-down in conjunction with the genetic associations only favors the conclusion that the proteins exist in a IFT-A/kinesin-2 architecture (lines 217-219) after Wg stimulation. We thank the reviewer for their positive statement about the quality of our data. We have adjusted the conclusion of this section as suggested by the reviewer (new sentence on lines 201-203).

Re 17) The authors insist on calling what looks to this referee to be membrane-enriched fraction as "cytoplasmic fraction". The cytoplasmic fraction is lost during the mincing in the PBS step (as explained in the method section), and the literature indicates cytoplasmic accumulation of beta-catenin upon Wnt3a stimulation (not seen in Fig 7j). I also do not find any clarification of this in the text or the legend. The authors can explain this or remove the panels for "cytoplasmic fraction" and replace them with total cell extract to show total beta-catenin levels for comparison with the nuclear fraction. For cytoplasmic fraction of beta-catenin the authors can use a hypotonic buffer or a saponin/digitonin-based buffer (see for example PMID: 29977599 and PMID: 25392450), but this is not required since the work is focused on nuclear localization of beta-catenin.

We have corrected this in the text and in the respective figures (now Figs. 8 and 9).

Other minor points:

Figure 4: Indicate in the figure if panels e-h are -Wg or +Wg.

The “Wg” description labels have been.

Line 1005: legend for figure 7 (change "mote" to "note").
This has been corrected, thank you for spotting this error.